# Probability Assessment of Static Overload in Wind Turbine Blade Bearings Considering Turbulence, Design, and Manufacturing Variability

Ashkan Rezaei[1] and Amir Rasekhi Nejad[1]

[1]Department of Marine Technology, Norwegian University of Science and Technology (NTNU), NO-7491, Trondheim, Norway

**Correspondence:** ashkan rezaei (ashkan.rezaei@ntnu.no)

**Abstract.** This study presents a probability assessment of static overload of a double-row, four-point contact ball blade bearing under ultimate limit state conditions. The methodology follows structural reliability principles applied to this specific limit state. The National Renewable Energy Laboratory 5 MW reference wind turbine is selected for the study, and a Monte Carlo simulation is used to assess static-overload reliability and estimate the probability of failure. The sensitivity of the probability of failure to uncertainties in turbulence intensity, material properties, and bearing dimensions is evaluated. Within the bounds examined, the blade bearing conformity has the largest effect on the probability of failure, and ball diameter is next. The probability of damage for case-study wind sites around the world is assessed, and it is observed that the probability of failure is higher in some cases than for wind conditions described by IEC 61400-1.

## 1 Introduction

Demand for wind turbines will grow significantly over the next decade, and wind turbines are one of the main keys to meeting the IEA Net Zero Emissions by 2050 (IEA, 2023). The efficient operation of these turbines depends on various components, with blade bearings, also named pitch bearings, being a critical element. Blade bearings facilitate the smooth rotation of the turbine blades, allowing them to capture the kinetic energy of the wind at different wind speeds without significantly increasing the structural loads on the turbine. Blade bearings serve as the connection point between the blades and the hub, allowing the blades to rotate around their axis. Although the entire pitch system assembly costs less than one percent of the wind turbine (Stehly et al., 2024), changing a blade bearing is costly due to the need to lower the blade with a large crane. The cost of replacement with a crane can reach up to $350,000 per week (Mishnaevsky Jr and Thomsen, 2020).

A common and distinguishing feature of the blade bearings is that they involve a rather slow oscillatory motion (Harris et al., 2009). This movement pattern differs from that of bearings in most other industrial applications, where bearings usually rotate continuously (Menck et al., 2020). Unlike most oscillating bearings, blade bearings perform stochastic oscillations rather than constant amplitudes back and forth (Stammler et al., 2024). They oscillate through only a few degrees, up to 20° (Keller and Guo, 2022) during normal power operation, depending on wind conditions and turbine sizes, and up to 90° in emergency feathering, rather than completing full revolutions. The limited rolling distance, long dwell periods at fixed pitch angles, and

frequent load reversals with high axial offsets concentrate stress cycles within a small contact zone on the raceways. Blade bearing failure consists of different damage modes, including rolling contact fatigue, core crushing, edge loading, ring fracture, rotational wear, fretting, false brinelling (Andreasen et al., 2022), and cage damage. Many of these failure mechanisms initiate in high-stress regions associated with maximum ball loads. While this study focuses on static overload, applied loads and bearing parameters that drive this mechanism are also relevant to other failure modes.

As part of the design and certification process, the static safety factor of the blade bearing under the ultimate limit state (ULS) must be assessed, as mentioned in IEC 61400-1 (2019); DNV-ST-0437 (2016); Harris et al. (2009); Germanischer Lloyd (2010); and Stammler et al. (2024).

Several studies have analyzed blade bearings. Among them, Menck et al. (2020) studied different lifetime calculation methods from the standards and guidelines and compared them to each other, highlighting differences in the methods and their results, and Schwack et al. (2016) compared different fatigue lifetime calculation methods and showed huge differences in the calculated fatigue lifetime for different approaches. There are not many studies analyzing the static safety factor in blade bearings. Keller and Guo (2022) studied the static load rating and safety factor of the blade bearing of a 1.5 MW wind turbine. They compared the ISO 76 (2006) methodology with the National Renewable Energy Laboratory's (NREL's) pitch and yaw bearing design guideline (DG03) (Harris et al., 2009) and concluded that the DG03 methodology is the same as the ISO 76 (2006) recommendation for applications subjected to shock loads. Rezaei et al. (2023) studied the blade bearing of the 5 MW NREL reference wind turbine and assessed the variation in blade bearing fatigue with shear power law exponent, turbulence intensity, and even resulting from each individual turbulent wind time series. In another work Rezaei and Nejad (2023), the fatigue life of the blade bearing is compared in different wind sites and compared with IEC-designed blade bearings. The results show that fatigue life at some wind sites is lower, even though their average wind speeds fall below the IEC 61400-1 (2019) category thresholds.

Even though it was shown that the results of wind sites are not the same as those of IECs, it is not clear how reliable the results are, as wind is a stochastic phenomenon. Haus et al. from 55+ GW of wind plant data show that blade bearings installed pre-2016 perform fairly well, only reaching a 10% replacement rate in 15 years. However, blade bearings installed post-2016 on larger wind turbines are projected to have a 10% replacement rate in only 7.5 years.

The design of blade bearings is governed by a combination of turbine-level and component-level standards. IEC 61400-1, Clause 9.8.4, requires that blade bearings demonstrate a minimum static safety factor against permanent deformation in the ultimate load cases. This requirement is based on limiting the local contact stress between the balls and raceways to a threshold value. DG03 expands on this by recommending a contact stress limit of 4200 MPa, based on ISO 76, and defining a methodology to evaluate loads, contact geometry, and bearing strength. DG03 is widely used in the industry but is expected to be superseded in the coming years by the proposed IEC 61400-18, which will standardize pitch and yaw bearing design. While IEC 61400-8 (2024) provides structural design guidance for nacelle and hub components, it does not explicitly include blade bearings within its scope. Nonetheless, it may become a valuable reference if adopted in future standards such as IEC 61400-18, just as IEC 61400-4 (2025) references it for gearbox structural components.

This paper analyzes the probability of static overload in a blade bearing at the ultimate limit state using a structural reliability framework, with a deeper focus on the effect of the wind conditions—particularly turbulence intensity—as well as uncertainties in bearing loads, material strength, and manufacturing tolerances. The goal is to quantify the probability that static contact stress exceeds a specified limit, not to assess all failure modes. While static overload is used here as a representative limit state due to its stress threshold (4200 MPa), the methodology is generalizable. Extreme loading, combined with uncertainty in geometry and material properties, is relevant to other critical failure modes—such as ring cracking—if the appropriate limit-state definition is available. A sensitivity analysis is also performed to identify which parameters (e.g., raceway conformity, ball diameter) most strongly influence the probability of static overload. Bearing static damage is considered a criterion for the ultimate limit state. ISO 76 (2006) states that experience shows that a total permanent deformation of 0.0001 of the ball diameter at the most heavily loaded contact without the subsequent operation being impaired. In the present study, we treat contact stresses that reach this deformation limit as attaining the ultimate limit state, i.e., stresses approaching the ISO 76 threshold are assumed to represent the onset of static failure and therefore increase the probability of bearing failure. This permanent deformation can cause stress concentrations of considerable magnitude and the formation of cavities in the raceways. These indentations, together with conditions of marginal lubrication, can also lead to surface-initiated fatigue damage (Harris and Kotzalas, 2006). Although some research showed that the slewing bearing can tolerate higher total permanent deformation while no core crushing occurred (Stammler et al., 2024), those results might be correct for a certain range of bearings with specific material and heat treatment (Lai et al., 2009). The current work assumes a contact stress of 4200 MPa as the criterion for static overload; however, there is a possibility that indentation and core crushing damage do not occur in all bearings at this level.

## 2  Case study: wind turbine and wind sites

### 2.1  Reference wind turbine

Load calculations were carried out using the NREL 5MW reference wind turbine by Jonkman et al. (2009). It is an offshore wind turbine, designed for wind class IEC IB. The turbine properties are displayed in Table 1.

**Table 1.** 5 MW NREL reference wind turbine specification (Jonkman et al., 2009)

| Wind Turbine | NREL 5 MW Reference Wind Turbine |
|---|---|
| Rating | 5 MW |
| Rotor Diameter | 126 m |
| Hub Height | 90 m |
| Drivetrain | High-speed, multiple-stage gearbox |
| Minimum and Rated Rotor Speed | 6.9 rpm, 12.1 rpm |
| Cut-In, Rated, Cut-Out Wind Speed | 3 m/s, 11.4 m/s, 25 m/s |
| Overhang, Shaft Tilt, Precone | 5 m, 5°, 2.5° |
| Rotor Mass | 110,000 kg |
| Nacelle Mass | 240,000 kg |
| Tower Mass | 347,460 kg |

## 2.2 Blade bearing

The blade-bearing model is considered from work by Rezaei et al. (2023). The bearing is a double-row, four-point contact ball bearing with a total of 250 balls. The general specification of the bearing is presented in Table 2. The details of the bearing specifications and dimensions can be accessed at Rezaei et al. (2023).

**Table 2.** Blade bearing main dimensions

| Parameter | Value | Description |
|---|---|---|
| $D_{pw}$ | 3558 | Bearing pitch circle diameter (mm) |
| $D$ | 75 | Ball diameter (mm) |
| $\alpha$ | 45 | Initial contact angle (°) |
| $Z$ | 125 | Number of balls per row |
| $i$ | 2 | Number of rows |
| $f_i$ | 0.53 | Inner raceway groove radius/$D$ |
| $f_o$ | 0.53 | Outer raceway groove radius/$D$ |

## 2.3 Wind sites

The wind regimes consist of IEC-category wind fields and wind sites. IEC-category wind fields consist of three wind speed classes: I, II, and III. Each wind speed class has four subclasses of A+, A, B, and C based on turbulence intensity. The basic parameters of the wind turbine classes are presented in Table 3 (IEC 61400-1, 2019). In the table $V_{ave}$ is the annual wind speed, $V_{ref}$ is the reference wind speed averaged over 10 minutes, and $I_{ref}$ is the reference value of the turbulence intensity.

**Table 3.** Basic parameters for IEC-category wind turbine (IEC 61400-1, 2019)

| Wind turbine class | I | II | III |
|---|---|---|---|
| $V_{ave}$ (m/s) | 10 | 8.5 | 7.5 |
| $V_{ref}$ (m/s) | 50 | 42.5 | 37.5 |
| A+, $I_{ref}$ | | 0.18 | |
| A, $I_{ref}$ | | 0.16 | |
| B, $I_{ref}$ | | 0.14 | |
| C, $I_{ref}$ | | 0.12 | |

The wind sites include 13 in Iran (SATBA, 2022), Pakistan (World Bank Group, 2023b), Vietnam (GIZ, 2023), Ethiopia (World Bank Group, 2023a), Denmark (Ørsted, 2022), and the United States of America (Jager and Andreas, 1996). The wind data includes the mean and standard deviation of 10-min wind speed. In order to account for the wind's seasonal effect, data needs to cover an entire year. The nominated wind sites cover a whole year's measurements. More information on the wind sites is presented at work by Rezaei and Nejad (2023).

In this study, the extreme turbulence model (ETM) was investigated. ETM is calculated according to IEC 61400-1 (2019) and prescribes rarer, higher-turbulence realizations that are used later in our ULS analysis. The extreme turbulence intensity at wind sites refers to the maximum turbulence intensity within each wind speed range. To connect the IEC references to the site data, the ETM turbulence intensity curve for IEC classes was computed and compared with the per-bin extreme TI observed at representative sites. Figure 1 shows that Aysha and Kebribeyah exhibit extreme TI values substantially above ETM for Class

IA (and, at some wind speeds, even above IA+), while Thanh Hai, Mil Nader, and Flatirons are closer to ETM.

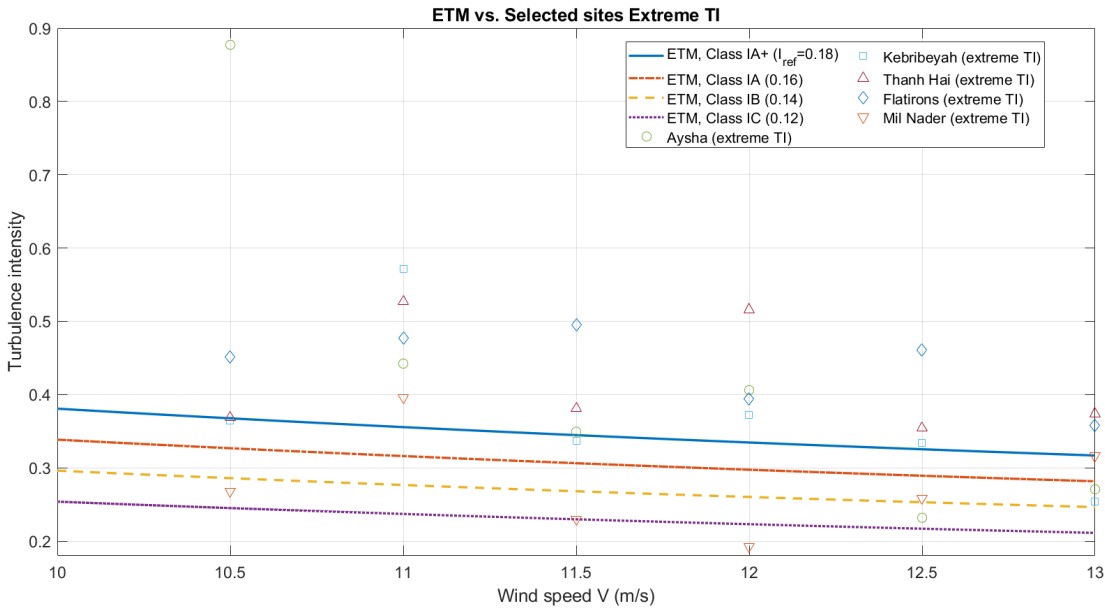

**Figure 1.** Comparison of turbulence intensity for IEC ETM and extreme conditions at selected wind sites

The extreme turbulence intensities at the wind sites are presented in Table 4. The effects of high turbulence intensity in some wind sites, such as Sujawal, Aysha, and Kebribeyah, are shown and discussed in the results section.

**Table 4.** Extreme turbulence intensity in wind sites

| Wind speed (m/s) | Khaf | Mil nader | Shourjeh | Bafrajerd | Moaleman | Sujawal | ThanhHai | Aysha | Gode | Kebribeyah | Tuluguled | Flatirons | Anholt |
|---|---|---|---|---|---|---|---|---|---|---|---|---|---|
| 10.5 | 0.3143 | 0.2682 | 0.4149 | 0.4825 | 0.2927 | 0.6513 | 0.3688 | 0.8773 | 0.2608 | 0.3641 | 0.3421 | 0.4517 | 0.2341 |
| 11 | 0.3412 | 0.3957 | 0.4616 | 0.3575 | 0.3109 | 0.171 | 0.5272 | 0.4422 | 0.261 | 0.5717 | 0.3427 | 0.4771 | 0.2193 |
| 11.5 | 0.361 | 0.2295 | 0.5302 | 0.3914 | 0.295 | 0.2135 | 0.3807 | 0.3493 | 0.2736 | 0.3364 | 0.3243 | 0.4953 | 0.2928 |
| 12 | 0.4366 | 0.1926 | 0.3572 | 0.3623 | 0.4885 | 0.166 | 0.5161 | 0.406 | 0.2406 | 0.3722 | 0.4477 | 0.3938 | 0.2467 |
| 12.5 | 0.3735 | 0.2587 | 0.3159 | 0.3683 | 0.2652 | 0.1659 | 0.3542 | 0.2318 | 0.237 | 0.3343 | 0.3629 | 0.4607 | 0.1704 |
| 13 | 0.3592 | 0.316 | 0.3188 | 0.3736 | 0.3125 | 0.1775 | 0.374 | 0.2707 | 0.2092 | 0.2537 | 0.1985 | 0.3577 | 0.1675 |

## 3 Methodology

### 3.1 Structural reliability

The study of structural reliability is concerned with calculating and predicting the probability of limit state violation for an engineered system at any stage during its life as defined by Melchers and Beck (2018). Nejad (2018) named the main aim of the structural reliability as an estimation of the failure probability by taking into account explicitly uncertainties of the load, load effect, and resistance.

Ultimate limit state reliability in the current work is based on a static safety factor. The important process used to calculate the ultimate limit state in this paper is illustrated in Fig. 2.

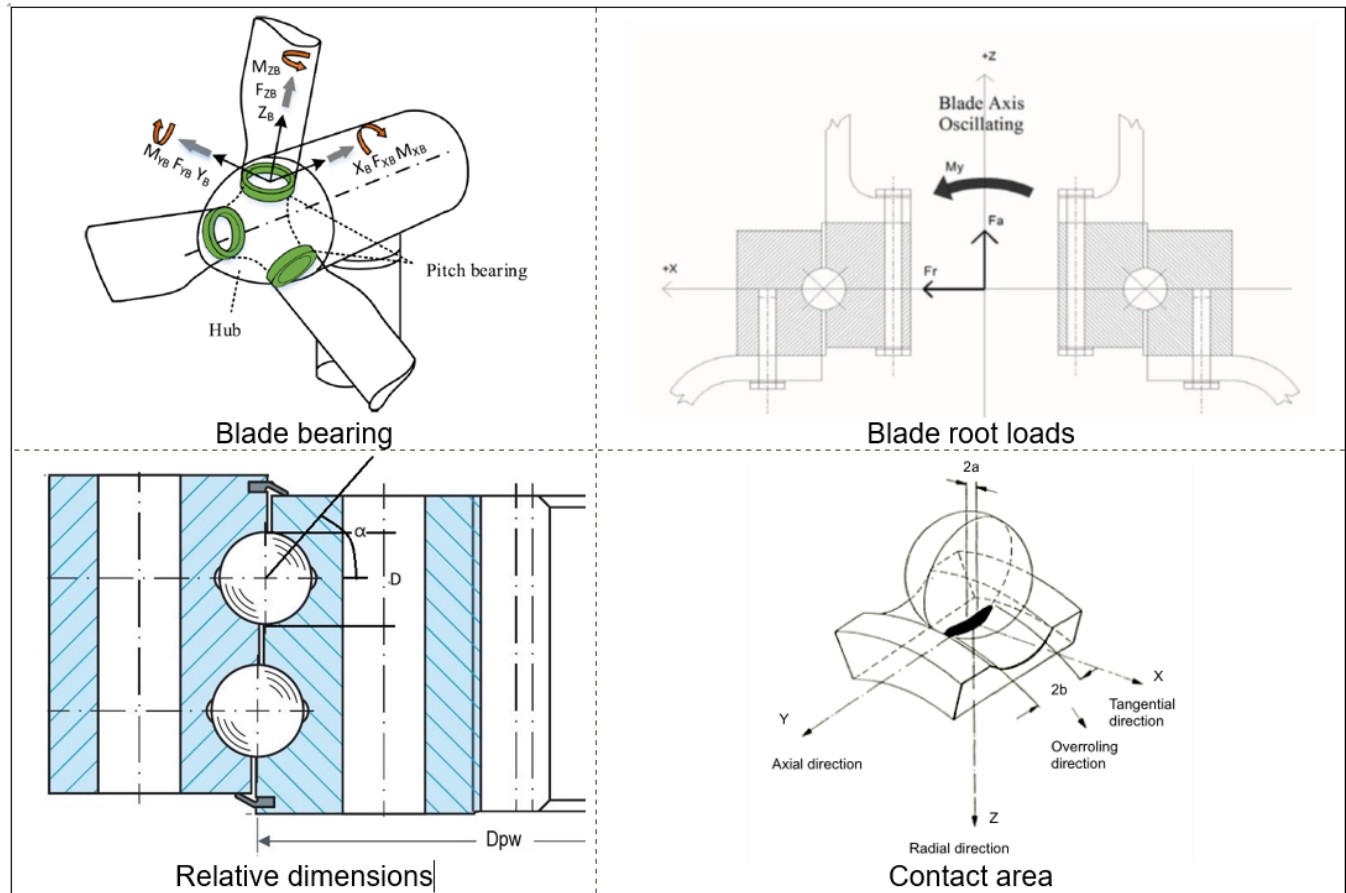

**Figure 2.** Illustration of the main processes used to calculate the static safety factor, images extracted from He et al. (2024); Harris et al. (2009); Rezaei et al. (2023); Solano-Alvarez (2015)

These processes started with simulating the wind turbine in different turbulence wind conditions. From the turbine responses, the loads on the blade bearing are derived. Different dimensions are studied, and material probability is considered. The contact area is calculated, and the safety factor is derived.

## 3.2 Safety factor and failure function

A safety factor is a measure used in engineering and design to provide a margin of safety for structures, materials, or systems under expected loads or conditions. It accounts for uncertainties in the design process, such as variations in material properties, manufacturing tolerances, unexpected loads, and potential degradation over time. In this study, the formula that is used for the failure function has the same form as the static calculation function and static safety factors.

For the reliability model, the static limit state is formulated with the Hertzian contact-stress criterion. The approach considers that bearing static damage occurs when the maximum Hertz contact stress exceeds the allowable Hertz contact stress. The DG03 uses a comparison of the maximum contact stress, $\sigma_{\max}$, in the limit load condition to the maximum allowable stress of 4,200 megapascals to define the static safety factor ($S_0$) (Harris et al., 2009; Stammler et al., 2024).

$$S_0 = (\frac{4200}{\sigma_{\max}})^3 \tag{1}$$

where the maximum contact stress, $\sigma_{max}$, is also expressed in megapascals (MPa) calculated as

$$\sigma_{max} = \frac{1.5 Q_{\max}}{\pi a b} \tag{2}$$

and the static safety factor can be rewritten as

$$S_0 = \left( \frac{\frac{4200 \pi a b}{1.5}}{Q_{\max}} \right)^3 \tag{3}$$

In equation 2, $\pi a b$ is the contact area, which is an ellipse having semi-major axis $a$ and semi-minor axis $b$, and $Q_{\max}$ is the maximum ball force. The maximum ball load is calculated from Stammler et al. (2024)

$$Q_{\max} = 0.55 \left( \frac{2 F_r}{Z \cos \alpha} + \frac{F_a}{Z \sin \alpha} + \frac{4.4 M}{D_{\mathrm{pw}} Z \sin \alpha} \right) \tag{4}$$

where $F_r$, $F_a$, and $M$ denote the applied radial, axial, and moment loads, respectively.

In Equation 3, the numerator and denominator are named $R$ and $S$, respectively. $a$ and $b$ in the $R$ are functions of the applied maximum ball load $Q_{\max}$; therefore, $R$ is implicitly a function of $S$. The failure function, $g_x$, is defined below, where $x$ are random variables and $n$ is the static safety ratio.

$$g_x(R,S) = (\frac{R}{S})^3 \leq n \tag{5}$$

The static safety ratio, $n$, determines the boundary of the damage. If the failure function value is equal to or smaller than the static safety ratio, the bearing is in a failure state; otherwise, the bearing is in a safe state. In a work by Keller and Guo (2022), it is recommended that the static safety ratio be greater than 1.5; however, Stammler et al. (2024) noted that it seems reasonable to refer to the limit of 1, such as IEC 61400-1 (2019). In order to compute the actual reliability and allow for comparisons, the static factor ratio equal to 1 is considered. Consequently, $g_x$ will become

$$g_x(R,S) = (\frac{R}{S})^3 \leq 1 \tag{6}$$

Taking the cube root of Equation 6 yields the failure function directly as

$$g_x(R,S) = R - S \leq 0 \tag{7}$$

The annual probability of failure, $P_f$, is then obtained from

$$P_f = P(g_x(R,S) \leq 0) = P(R - S \leq 0) \tag{8}$$

To compute $P_f$, different methods can be used, including first-order and second-order reliability methods (FORM and SORM), as well as the Monte Carlo method (Ditlevsen and Madsen, 2007). In this study, the Monte Carlo simulation method is employed to estimate the probability of failure. By the Monte Carlo simulation method, a suitably large sample of typical load configurations is simulated from the probabilistic action model. This load configuration sample gives a corresponding sample of load effects at different points of the bearing, and from this sample, the probability distributions of the load effects can be estimated. By using these probability distributions, extreme value studies can next be made. In this sense, the probabilistic model uses typical load configurations in its solution procedure and not difficult choices of "extreme" load configurations as they are used in the deterministic model (Ditlevsen and Madsen, 2007).

The randomness of the failure function arises from different aspects. Uncertainties in material, forces, and models are some of the main ones. These sources of randomness can appear in the $R$ representation of the load-capacity term and $S$ the representation of the applied maximum ball load. The randomness in $R$ and $S$ in the current work originated from uncertainty in the materials, dimensions, wind turbulence intensity, and simulation model in ball forces. Figure 3 presents a systematic approach for the static overload reliability assessment of a blade bearing.

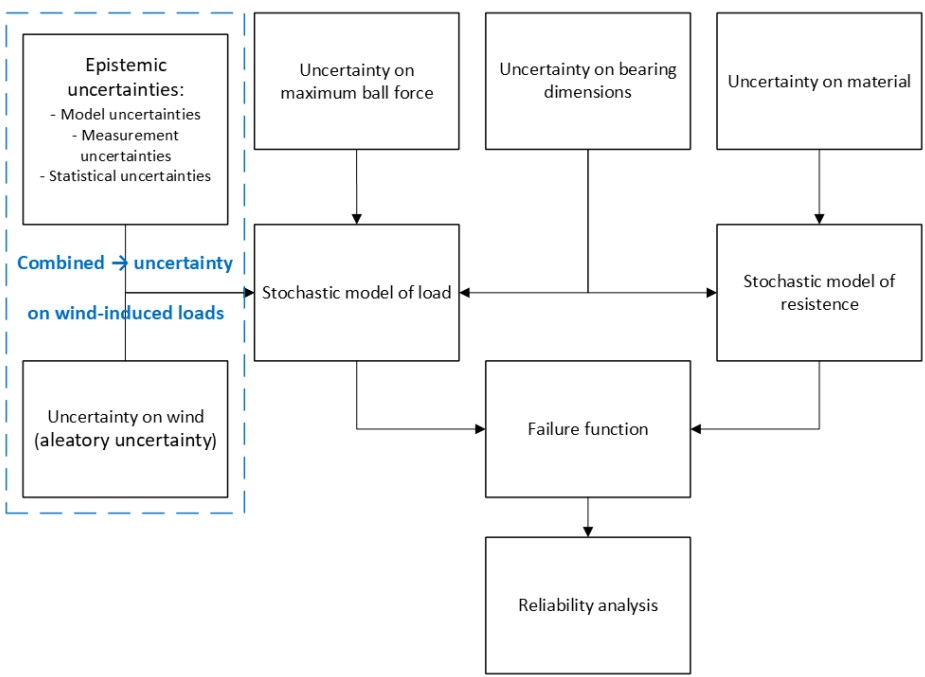

**Figure 3.** Flowchart of reliability assessment in a blade bearing for ultimate limit state

The dashed box represents uncertainty on loads that have distinct uncertainty sources -(i) external wind variability and (ii) Epistemic uncertainty arises from model, measurement, and stochastic uncertainties, which are sampled separately and then combined to generate the stochastic wind-induced loads. The uncertainties of the measurement and statistics are not considered in this study. Uncertainty in material leads to uncertainty in resistance, while uncertainty in dimension leads to both uncertainty

in stress and resistance. The formula for resistance extracted from Equation 9 with consideration of uncertainty in material and dimension is presented below,

$$R = \left( \frac{4200\chi_m\pi a(\chi_d)b(\chi_d)}{1.5} \right) \qquad (9)$$

where $\chi_m$ and $\chi_d$ are uncertainties in material and dimension, respectively. The uncertainties in blade bearing design dimensions, such as pitch diameter, ball diameter, contact angle, and groove conformity, affect the dimensions of the contact area, $a$ and $b$. It should be noted that $a$ and $b$ are, in addition to the uncertainty in dimension, functions of uncertainty in the loads and the maximum ball forces $Q_{\max}$. In every Monte-Carlo realization, therefore, $Q_{\max}$ is computed first, then evaluated $a$, $b$, and finally $R$, ensuring that the dependency R(S) is fully captured. The formula that represents $S$, presented below.

$$S = (Q_{\max}\chi_f) \qquad (10)$$

where $\chi_f$ is uncertainty in maximum ball force. $Q_{\max}$ contains external loads and bearing dimensions, which in fact are uncertain parameters. Therefore, in its nature, it consists of uncertainty.

The probability of failure including all uncertainties will then be

$$P_f = P(\left( \frac{4200\chi_m\pi a(\chi_d)b(\chi_d)}{1.5} \right) - (Q_{\max}\chi_f) \leq 0) \qquad (11)$$

### 3.2.1 Uncertainty in material

The strength of bearings is not a deterministic value. ISO 76 (2006) recommends a Hertz contact stress of 4200 MPa for ball bearings, which is equivalent to a total permanent deformation of 0.0001 of the rolling element diameter. Assessing the uncertainty of the material's strength requires extensive testing. Shimizu et al. (2010) presented the mechanical properties of the AISI 52100 bearing steel. They showed that tensile strength and Rockwell C hardness have a normal distribution with a standard deviation of 5.7% and 1.1%, respectively. Lai et al. (2009) presented a model for plastic indentation, and they tested it on 42CrMo4 steel. Their model predicted that the contact pressure for causing plastic indentation of $10^{-4}D$ in the through-hard raceway is 4260 MPa, as well as good validation results. In the extended work, Lai (2011) predicted the contact pressure to be 4270 MPa. Lewis et al. (2015) assessed 9 different sources, including 7 bearing manufacturers, and allowable peak Hertzian pressures of 4270 MPa and 3962 MPa for SAE 52100 steel and AISI 440C steel, respectively, based on the mean minimum hardness. Wang and Zhang (2022) performed a reliability analysis on an angular contact ball bearing. They considered allowable yield stress as a strength parameter with a 5% standard deviation. In another work, density and modulus of elasticity in work by Cheng et al. (2020) are considered a normal distribution with a 5% standard deviation. In their sensitivity analysis, they concluded that the material has the highest reliability sensitivity. Imdad et al. (2024) studied 42CrMo4 steel that was submitted to different heat treatments and hardness levels. The hardness level in different heat treatments has a deviation between 2% and 6.5%. The current work considered a normal distribution with a standard deviation of 5.7% for the uncertainty of material, $\chi_m$.

### 3.2.2 Uncertainty in dimensions

Blade bearing dimensions affect the loads and resistance in the failure function. Wang and Zhang (2022) performed a sensitivity analysis on four parameters of the bearing, including ball pitch diameter, ball diameter, and inner and outer raceway groove curvature with normal distributions and 0.5% standard deviations. They concluded that, regarding the bearing geometry, the ball diameter has the highest effect on reliability to prevent plastic deformation. In Cheng et al. (2020)'s sensitivity analysis of an angular contact ball bearing, the free contact angle and the inner and outer raceway curvatures were treated as random parameters, along with the ball diameter and groove curvatures. All of their random parameters had 0.5% standard deviations. In a reliability and sensitivity analysis of spherical roller bearings, Wang et al. (2020) considered 4%, 2.2%, and 0.5% standard deviation for roller diameter, pitch circle diameter, and radial clearance, respectively. The effect of dimensions on the reliability of the bearing is presented in the sensitivity analysis section. It should be noted that each dimension was analyzed independently by the relevant distribution.

### 3.2.3 Uncertainty in loads

Wang and Zhang (2022), in their work, considered normal distributions with a 5% standard deviation for axial and radial forces. The same consideration at work by Cheng et al. (2020) has been seen. In contrast, Wang et al. (2020) considered a standard deviation of 2.5% for the radial load.

The uncertainty of the load at the blade bearing originates mainly from turbulence acting on the wind turbine, and the turbulence has a great contribution to both the safety factor and the fatigue life of the blade bearing, as shown in the work by Rezaei et al. (2023). Different realizations of the turbulence, called "seed number," produce a Gaussian distribution of TI in the longitudinal wind component due to spatial coherence (Jonkman, 2009).

Each simulation with a specific seed number leads to a time series of distributions of the loads in the balls, while the extreme ball load can be obtained from these series. Different random seed number simulations result in a series of extreme ball loads in the blade bearing while the turbulence intensity is constant. These extreme loads form a probability distribution function. It is important to assign a proper probability distribution function to these extreme loads.

In this study, the following probability distribution functions were considered: the Generalized Extreme Value, Gamma, Inverse Gaussian, Kernel, Lognormal, Nakagami, Rician, and Weibull distributions. In Table 5, the probability density function (PDF) of the nominated distribution function is presented, where $x$ is a random variable. More information on the equations and parameter definitions of generalized extreme value, Gamma, Kernel, and Weibull is referred to in Shi et al. (2021). The parameters of inverse Gaussian, lognormal, and Nakagami are referred to in Alavi et al. (2016). Rician parameters are referred to in Yu et al. (2019).

**Table 5.** PDFs of nominated distribution function (Alavi et al., 2016; Shi et al., 2021; Yu et al., 2019)

| Distribution function | PDF |
| --- | --- |
| Generalized Extreme Value (GEV) | $f(x) = \frac{1}{\alpha}[1 - \frac{k}{\alpha}(x - \mu)]^{\frac{1}{k}-1} - e^{-[1-\frac{k}{\alpha}(x-\mu)]^{\frac{1}{k}}}$ |
| Gamma (Gam) | $f(x) = \frac{\alpha^k}{\Gamma(k)} x^{k-1} e^{-\alpha x}$ |
| Inverse Gaussian (IG) | $f(x) = \sqrt{\frac{\lambda}{2\pi x^3}} e^{-\frac{\lambda}{2\mu^2 x}(x - \mu^2)}$ |
| Kernel (Ker) | $f(\alpha) = \frac{1}{nh} \sum_{i=1}^{n} K(\alpha), \quad \alpha = \frac{x - x_i}{h}$ |
| Lognormal (LN) | $f(x) = \frac{1}{x\sigma\sqrt{2\pi}} e^{-\frac{1}{2}[\frac{ln(x)-\mu}{\sigma}]^2}$ |
| Nakagami (Nak) | $f(x) = \frac{2m^m}{\Gamma(m)\Omega^m} x^{2m-1} e^{-\frac{m}{\Omega}x^2}$ |
| Rician (Ric) | $f(x) = \frac{x}{a^2} e^{-\frac{x^2+b^2}{2a^2}} I_0(\frac{bx}{a^2})$ |
| Weibull (Wbl) | $f(x) = \frac{k}{\alpha}(\frac{x}{\alpha})^{k-1} e^{-(\frac{x}{\alpha})^k}$ |

In this study, the parameters were calculated by the maximum likelihood estimator (Bain and Antle, 1967) using MATLAB software, and to assess the performance and goodness-of-fit (GoF) of the distribution functions, the coefficient of efficiency method (CE) has been applied. CE is intended to range from zero to one, but negative scores are also permitted. The maximum

positive score of one represents a perfect model; a value of zero indicates that the model is no better than a one-parameter "no knowledge" model in which the forecast is the mean of the observed series at all time steps; negative scores are unbounded; and a negative value indicates that the model is performing worse than a "no knowledge" model (Dawson et al., 2007). The CE indicator is one minus the ratio of the sum square error to the statistical variance of the observed dataset about the mean of the observed dataset.

$$CE = 1 - \frac{\sum (Q_i - \hat{Q}_i)^2}{\sum (Q_i - \bar{Q})^2} \tag{12}$$

$Q_i$ is observed data at level $i$, $\hat{Q}_i$ is estimated data at level $i$, and $\bar{Q}$ is the mean of observed data.

Different seed numbers were studied to create a distribution function. According to IEC 61400-1 (2019), in ultimate strength analysis, 15 different simulations are necessary for each wind speed from $(V_r - 2m/s)$ to cut-out, and six simulations are necessary for each wind speed below $(V_r - 2m/s)$ and $V_r$ is the rated wind speed that is 11.4 m/s according to Table 1.

However, for generating coherent turbulent structures, using more than 30 different random seeds for a specific set of boundary conditions is recommended by Jonkman (2009). This study covers a wide range of seed numbers. The generalized extreme value distribution is selected to model the maximum load distribution. The reason behind is presented in the results section.

### 3.2.4   Uncertainty in the maximum ball force

The uncertainty in the maximum ball forces arises from the distribution of the forces inside bearings. The flexibility of the
240 bearings and connecting components, hub and blade, can affect the load distributions inside the bearing, as was shown in works by Menck et al. (2020) and Rezaei et al. (2024). In addition, the results of the maximum ball force equation are not necessarily conservative. It can overestimate or underestimate the actual loads (Stammler et al., 2024).

Menck et al. (2020) developed a finite element model (FEM) and calculated the bearing load distributions. Moreover, Graß-mann et al. (2023) validated the finite element model with extensive experimental data on the blade bearing. Furthermore,

Rezaei et al. (2024) compared the load distribution from the finite element model and multi-body simulations (MBS) for NREL 5MW and IWES 7.5 MW wind turbines. The average of the maximum ball force differences between MBS and FEM was 10.8%. In another work by Leupold et al. (2021), load distributions inside the bearing for two different conditions of finite element and multi-body simulation were studied, and the average error was 6.5%.

    The maximum ball forces and load distributions in the work by Rezaei et al. (2024) were recalculated with the maximum
ball force equation. The differences between the maximum forces from FEM and the maximum ball equation have a mean and standard deviation of 13% and 6.5%, respectively, which is considered an uncertainty of maximum ball forces, $\chi_f$.

    The distributions and the mean and standard deviation of model uncertainties based on the above discussions are summarized in Table 6.

**Table 6.** Uncertainty distributions

| Uncertainty | Distribution | Mean | St. dev. |
|---|---|---|---|
| $\chi_f$ | Normal | 1.13 | 0.065 |
| $\chi_m$ | Normal | 1 | 0.057 |
| $\chi_d$ | Normal | 1 | 0.005 |
| $Q_{\max}$ | GEV | Distribution depends on each wind condition | |

### 3.3    Description of DLC

It is observed that design load case (DLC) 1.3 of IEC 61400-1, around rated wind speed, has the largest effect on the load of the blade bearings (Rezaei et al., 2023); therefore, it is considered a nominated load case.

    The DLC 1.3 uses an extreme turbulence model. The DLC covered mean wind speeds from 10 to 13 m/s with an interval of 0.5 m/s. The simulations last 700 seconds, and the results of the first hundred seconds are not considered. In all DLCs, wind shear is considered according to the sites or related standard conditions.

## 4    Results and discussion

### 4.1    Probability distribution function

By increasing the seed number, a bigger number of realizations is created, and the accuracy of the result is higher, but on the other hand, the simulation time will increase. Seed numbers from 15 to 3000 were studied in onshore wind conditions in IEC category IA. The results of the probability density function and exceedance probability of the fitted distributions for border
seed numbers 15 and 3000 are depicted in Fig. 4 and Fig. 5.

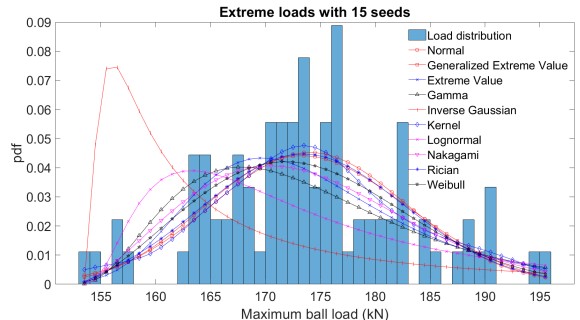
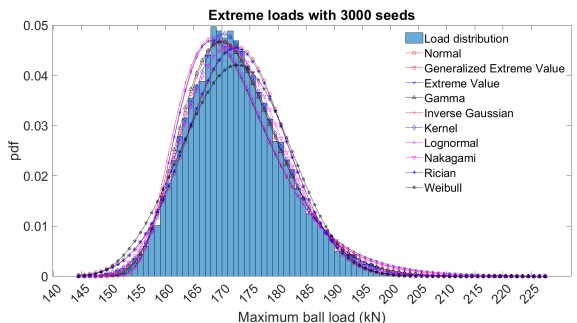

**Figure 4.** Annual probability density function of different distribution functions in 15 seeds (left) and 3000 seeds (right)

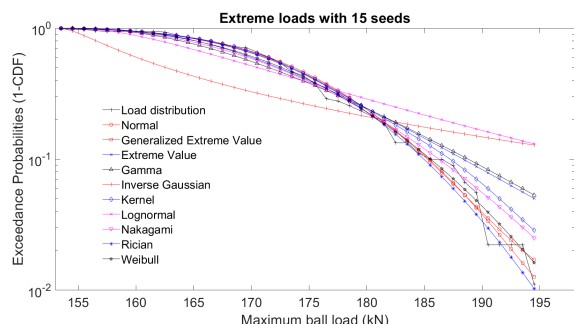
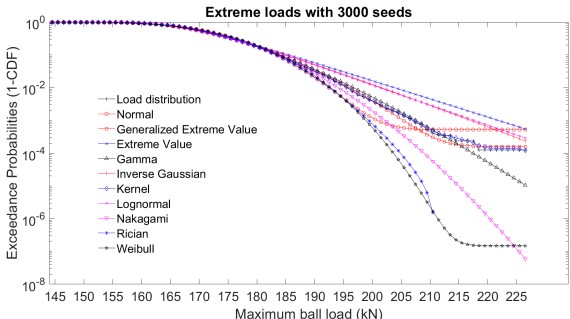

**Figure 5.** The exceedance probabilities of different distribution functions in 15 seeds (left) and 3000 seeds (right)

CE indicator results in different seed numbers with nominated probability distribution functions are presented in Table 7

**Table 7.** Coefficient of efficiencies of the nominated probability distribution functions in different seed numbers

| PDF | CE | | | | | | | | | | | | | | |
|-----|------|------|------|------|------|------|------|------|------|------|------|------|------|------|------|
| | 15 | 30 | 50 | 75 | 100 | 150 | 200 | 300 | 600 | 900 | 1200 | 1500 | 1800 | 2400 | 3000 |
| Nor | 0.464 | 0.677 | 0.793 | 0.753 | 0.871 | 0.872 | 0.944 | 0.942 | 0.961 | 0.961 | 0.972 | 0.971 | 0.977 | 0.977 | 0.977 |
| GEV | 0.461 | 0.672 | 0.858 | 0.807 | 0.921 | 0.914 | 0.974 | 0.972 | 0.982 | 0.982 | 0.993 | 0.994 | 0.994 | 0.995 | 0.996 |
| EV | 0.409 | 0.587 | 0.855 | 0.814 | 0.911 | 0.906 | 0.951 | 0.949 | 0.953 | 0.953 | 0.970 | 0.975 | 0.970 | 0.971 | 0.976 |
| Gam | 0.307 | 0.517 | 0.846 | 0.812 | 0.912 | 0.913 | 0.958 | 0.963 | 0.970 | 0.970 | 0.985 | 0.993 | 0.984 | 0.988 | 0.995 |
| IG | -1.218 | -0.685 | 0.614 | -0.169 | 0.802 | 0.791 | 0.818 | 0.805 | 0.860 | 0.860 | 0.911 | 0.965 | 0.904 | 0.932 | 0.968 |
| Ker | 0.510 | 0.703 | 0.867 | 0.836 | 0.928 | 0.920 | 0.977 | 0.975 | 0.989 | 0.989 | 0.997 | 0.997 | 0.997 | 0.999 | 0.999 |
| LN | 0.023 | 0.262 | 0.767 | 0.691 | 0.865 | 0.883 | 0.899 | 0.921 | 0.927 | 0.927 | 0.949 | 0.977 | 0.946 | 0.957 | 0.978 |
| Nak | 0.392 | 0.613 | 0.852 | 0.822 | 0.913 | 0.909 | 0.969 | 0.966 | 0.978 | 0.978 | 0.990 | 0.990 | 0.992 | 0.993 | 0.994 |
| Ric | 0.459 | 0.667 | 0.834 | 0.791 | 0.888 | 0.881 | 0.953 | 0.948 | 0.965 | 0.965 | 0.977 | 0.974 | 0.981 | 0.980 | 0.979 |
| Wbl | 0.429 | 0.641 | 0.846 | 0.812 | 0.897 | 0.883 | 0.955 | 0.939 | 0.955 | 0.955 | 0.971 | 0.957 | 0.975 | 0.971 | 0.964 |

The results show that the kernel and generalized extreme value distributions perform best in modeling the extreme load distribution. This dominance started with 150 seeds. The kernel estimator does not have a closed formula, and the generalized extreme value is considered a distribution for modeling the effect of seed number on the distribution of extreme loads. The results also indicate that a low number of seed values cannot accurately represent the true variety of the extreme loads.

Changes in the mean of the GEV function due to the seed number are plotted in Fig. 6.

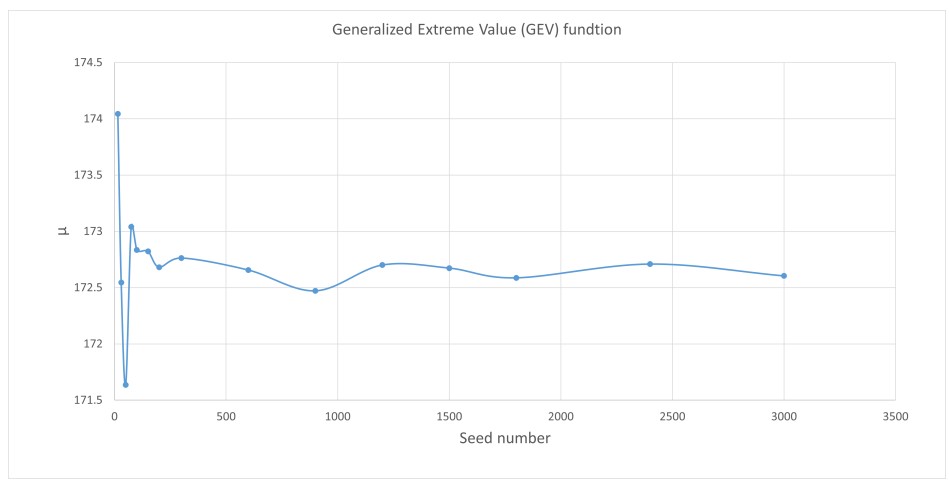

**Figure 6.** Changes of mean of generalized extreme value distribution due to seed numbers

It is very time-consuming to consider the high number of random seeds in the simulations. In addition, changes in the GEV parameters in 300 seeds and more are less than 3%. The 300 seed number is the value that is considered for the random seed number in the rest of the results.

## 4.2  Sensitivity analysis

The probability of failure in the bearing with variation in the ball diameter, pitch circle diameter, conformity, and contact angle is studied. The onshore wind field with an extreme turbulence intensity grade of IA according to IEC 61400-1, DLC 1.3, is considered. $10^8$ samples were considered in the simulation with the Monte Carlo method, and this process was repeated 20 times that forming a cluster.

### 4.2.1  Ball diameter

The nominal size of the ball diameter is 75 mm. Although the balls are usually manufactured and sorted in a batch with fine tolerances in diameters, it is assumed that the ball diameter can change from fine to very coarse machining according to ISO 2768-1 (1989). In every analysis, the balls' diameters are assumed to be the same, and a range of diameters was studied. However, the extreme tolerances are not realistic; they can help to observe the trend of changes in reliability. The assumption leads to a 0.15 to 1.5 mm variation in the ball diameter. The failure probability of the bearing with different ball diameters is shown in Fig. 7a. The vertical error bar refers to the maximum and minimum values in each cluster. Increasing the ball diameter increases the reliability of the bearing, which is in good agreement with the results from Wang and Zhang (2022); however, the reliability decreases more sharply in their analysis.

### 4.2.2 Pitch circle diameter

The nominal size of the pitch circle diameter is 3558 mm. It is assumed that the pitch circle diameter can change from medium (2 mm) to very coarse (8 mm) machining, according to ISO 2768-1 (1989). In order to observe the wider range of diameters, diameters of 3540 and 3576 mm were added to the study. The failure probability of the bearing with different pitch circle diameters is shown in Fig. 7b. The results show that the pitch circle diameter does not have a significant effect on the $P_f$.

### 4.2.3 Raceway conformity

Raceway conformity is the dimensional relationship between the radius of the raceway and the diameter of the ball. The nominal size of the conformity in the current study is 0.53. Bearing manufacturers recommend a value between 0.510 and 0.543 for this ratio (Daidié et al., 2008). In this study, the conformity between 0.515 and 0.545 is studied. The failure probability of the bearing with different raceway conformities is shown in Fig. 7c. The results of the $P_f$ show that with an increase in conformity, reliability sharply decreases. Wang and Zhang (2022) reached the same conclusion, but the decrease in reliability was not as sharp as the results presented. Wang et al. (2016) obtained similar results regarding the maximum Hertzian contact stress in their study on angular contact ball bearings. In order to understand how much the manufacturing of the ball and raceway can affect the reliability of the bearing, it is assumed that the ball and raceways have a fine degree of manufacturing according to ISO 2768-1 (1989), where in our study the tolerance would be 0.15 mm and the extreme values for conformity would be 0.527 and 0.533. The extreme values for the raceway conformity by this assumption are shown with vertical lines in Fig. 7c. It is observed that raceway conformity has the dominant effect on blade bearing damage in ULS. The result shows that with small changes in groove conformity, the probability of failure increases or decreases significantly. Consequently, the uncertainty of the raceway conformity with normal distribution with a standard deviation of 0.5% for the uncertainty of dimension, $\chi_d$, is considered.

### 4.2.4 Contact angle

The nominal size of the initial contact angle is $45°$. The initial contact angle in this study referred to the nominal contact angle, which is in a load-free condition. The contact angle from $25°$ to $65°$ is studied. The failure probability of the bearing with different contact angles is shown in Fig. 7d.

The results show that the probability of failure decreases as the contact angle increases. Cheng et al. (2020) got the same results in the shear stress in their research for angular contact ball bearings.

The initial contact angle, pitch circle diameter, and ball diameter do not have a significant effect on failure probability and are not considered in the paper as an uncertainty variable regarding dimension.

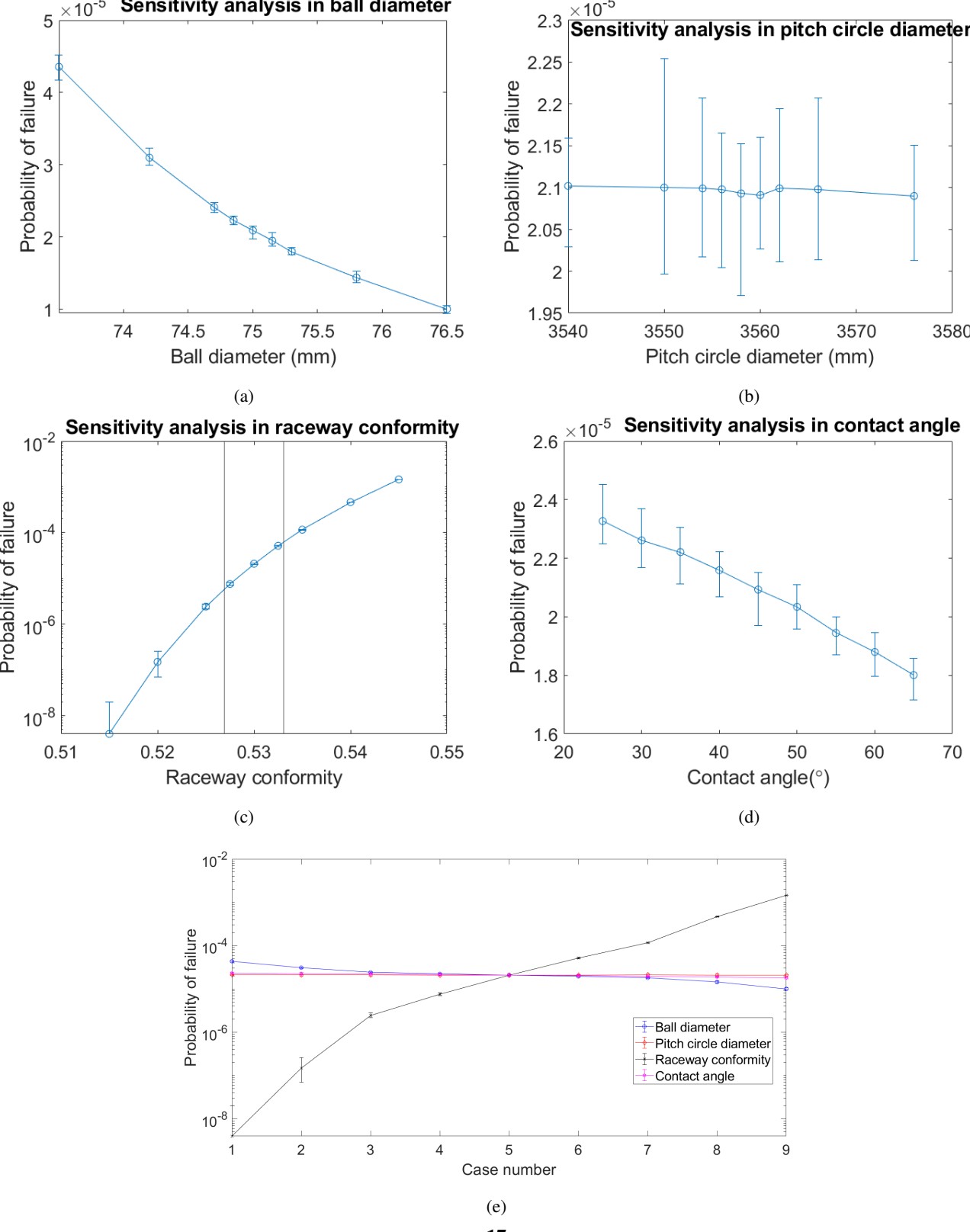

**Figure 7.** $P_f$ with different (a) ball diameter, (b) pitch circle diameter, (c) raceway conformity, (d) contact angle, and (e) summary of a to d

## 4.3 IEC wind conditions

The IEC wind categories I, II, and III in the turbulence intensity of A, B, and C at onshore and offshore conditions are studied. Simulations with different numbers of samples were performed. The result of the $P_f$ in IEC class I is depicted in Fig. 8.

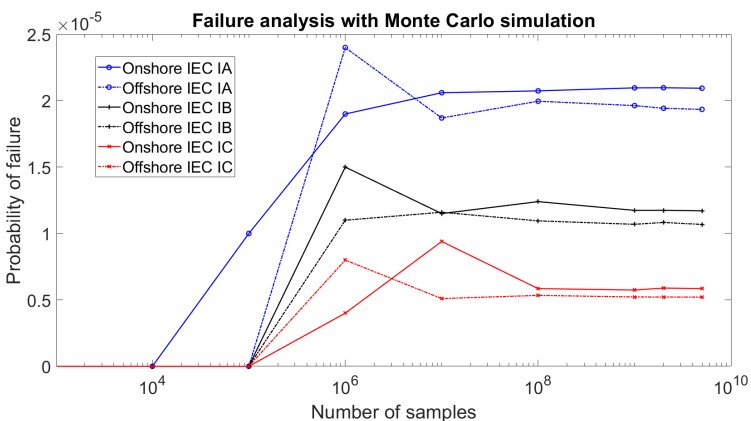

**Figure 8.** ULS probability of failure of IEC class I wind conditions in different sample numbers in Monte Carlo simulation

As the results show, the $P_f$ converges after $10^7$ samples in all wind conditions. To account for a wide range of samples, $10^8$ samples were considered in the simulation with the Monte Carlo method, which was repeated 20 times (20 clusters of $10^8$ samples). The $P_f$ is the average of 20 clusters. The IEC wind condition probability of failure based on the reference parameters of the bearing stated in Table 2 is illustrated in Fig. 9.

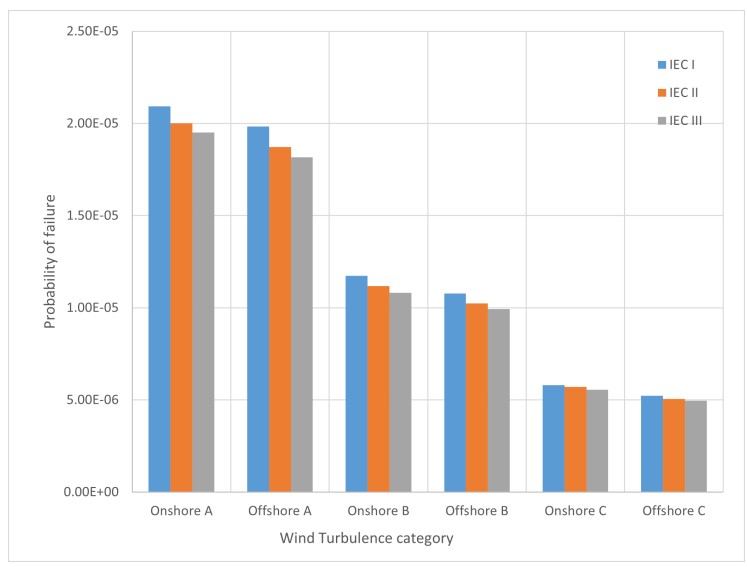

**Figure 9.** ULS annual probability of failure of IEC wind conditions

IEC onshore A class IA has the highest probability of failure, and the overall probabilities of failures of IEC wind config-
325 urations are in the order of $10^{-5}$. By increasing the annual mean wind speed, reliability decreases; however, the turbulence
intensity has a more significant effect, and reliability decreases when the turbulence intensity increases. While the $P_f$ mean
value varies between $2.09 \times 10^{-5}$ and $4.95 \times 10^{-6}$, the standard deviation of the clusters varies between $5.56 \times 10^{-7}$ and
$1.9 \times 10^{-7}$. The variance of the results is too small, and it indicates that the clusters are closer together, suggesting less diver-
sity and more consistency. IEC 61400-1 (2019); IEC 61400-8 (2024) set a target value for the nominal failure probability for
structural design for extreme and fatigue failure modes for a reference period of one year is $5 \times 10^{-4}$ for component class 2.
Component class 2 is "safe-life" structural components whose failure may lead to the failure of a major part of a wind turbine
(IEC 61400-8, 2024). All the wind configurations have a lower failure probability than the target value.

IEC 61400-1 (2019) recommends 15 seed numbers in the ultimate analysis. It is shown that 15 seed numbers cannot represent
the behavior of the probability distribution of the extreme loads. To investigate further, the load index is introduced. The load
index is the ratio of the extreme ball loads in 300 seed numbers to the extreme ball loads in 15 seed numbers. The load index
results for the IEC wind categories are plotted in Fig. 10. The results show that the extreme load calculation with 15 seeds has
an error between 2% and 11%. This exercise is referred to as a code–site comparison.

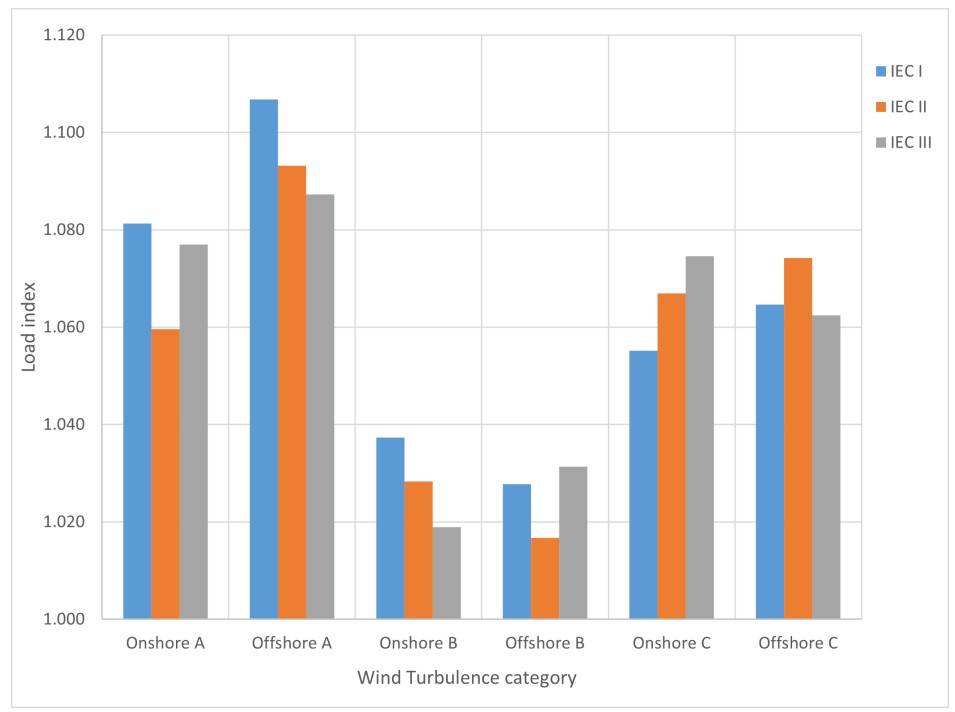

**Figure 10.** Load index of IEC wind conditions

## 4.4   Wind sites

The wind sites that were introduced previously were studied. The annual probability of failure for the nominated wind sites is
illustrated in Fig. 11. For comparison, the maximum $P_f$ value among the IEC wind conditions is also included in the figure.

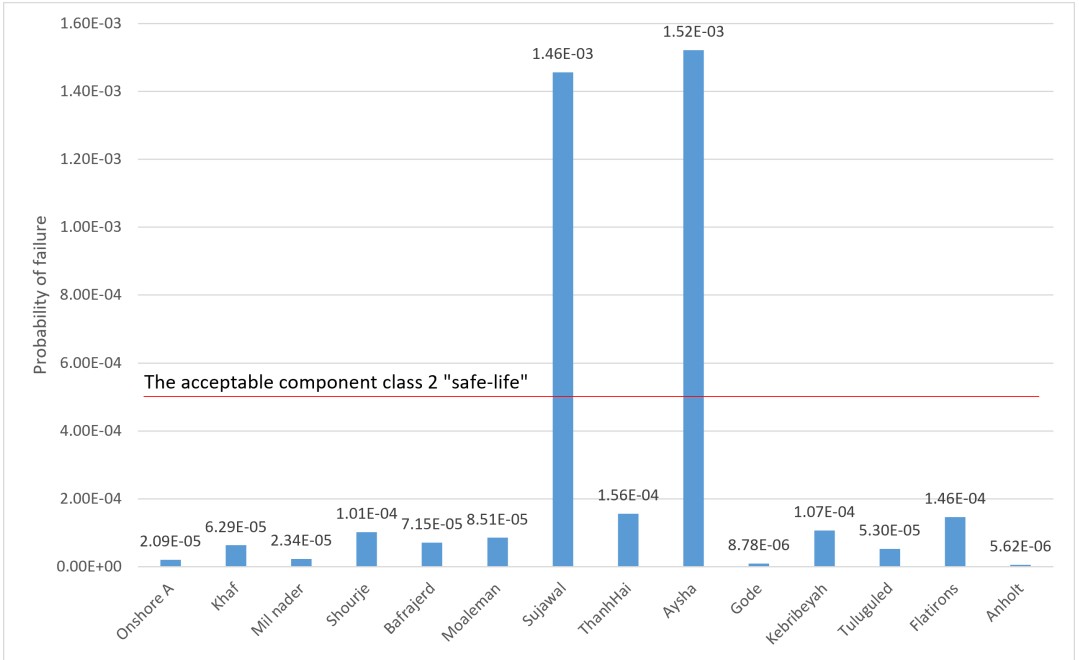

**Figure 11.** ULS annual probability of failure of nominated wind sites

The results show that most wind sites exhibit a higher probability of failure in the blade bearing under ULS conditions than the IEC categories. The reliability at the Sujawal and Aysha wind sites is far lower than that of the IECs. In addition, these two sites have higher failure probabilities than the failure target value for component class 2 (IEC 61400-8, 2024). The standard deviation of the clusters varies between $4.82 \times 10^{-6}$ and $2.33 \times 10^{-7}$, and the results of the clusters are consistent. These two
wind sites have annual wind speeds between 7.5 and 8.5 and are categorized in the IEC II class, while their $P_f$ is higher than the IEC I class wind sites. These high $P_f$ are the result of high turbulence intensity, as addressed in Table 4. The load index results for the wind sites are plotted in Fig. 12.

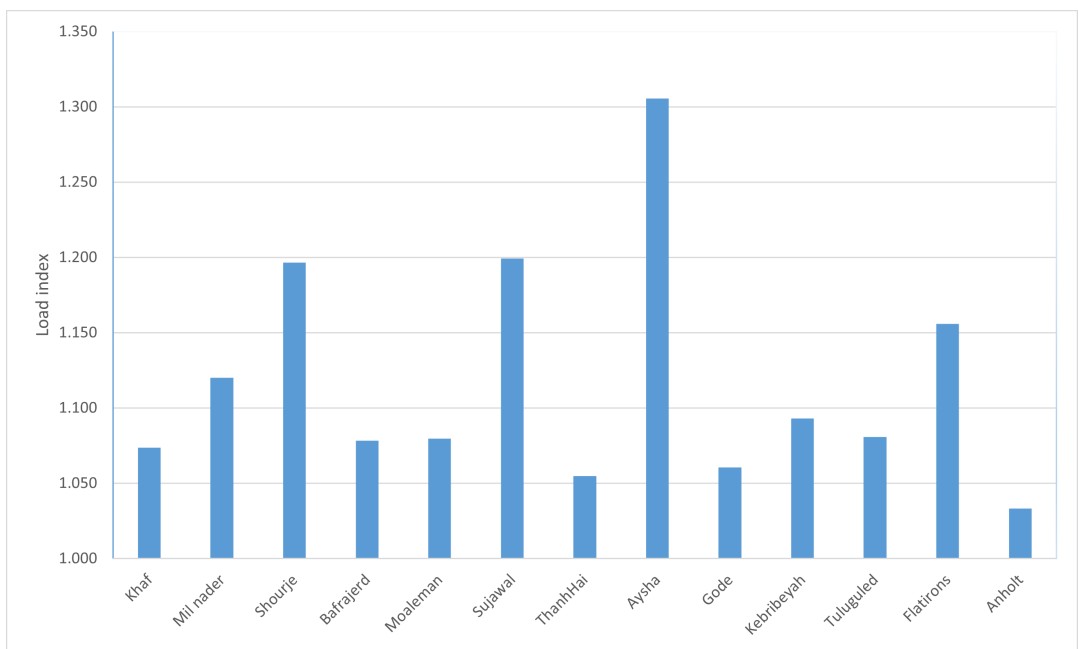

**Figure 12.** Load index of wind sites

The results show that the extreme load calculation with 15 seeds has an error between 3% and 30%. This observation has important implications for reliability analysis and simulation practice. It shows that using only 15 seeds, as typically done in standard simulations, can lead to a non-negligible underestimation of extreme ball loads, especially in complex wind site conditions. The load index provides a simple and effective measure to quantify this underestimation. It can help practitioners evaluate whether their simulation setup sufficiently captures the load extremes, and when limited seed numbers are used, the load index can offer a basis for applying correction factors to improve the accuracy of failure probability estimates. Additionally, the variation in load index across different wind sites emphasizes the need for site-specific assessment when evaluating blade bearing reliability.

## 5   Conclusions

The presented work evaluates the probability of static overload in a double-row, four-point contact ball blade bearing under ultimate limit state conditions. The analysis is based on a structural reliability approach, quantifying the likelihood of permanent deformation and the static safety factor, which is the ratio between the maximum permissible Hertzian contact stress and the maximum contact stress. The NREL 5 MW reference wind turbine is considered with the extreme turbulence wind model in the design load case. The Monte Carlo method was used to calculate the probability of failure. It is shown that the generalized extreme value is a suitable distribution to simulate the probability of the extreme ball load distribution. Increasing the number

of seeds in the turbulence simulation improves the accuracy of the estimated probability distribution. It is observed that by considering 15 seed numbers, as proposed in the standards and guidelines, the distribution of the loads is not represented.

The probability of failures of the blade bearing regarding variations of four main dimensions was studied. Raceway conformity in this aspect has the highest contribution to the probability of static overload of the blade bearing, and ball diameter is next, and strictly controlling these parameters can lead to higher reliability in the bearing regarding the ultimate limit state.

The $P_f$ results of different IEC wind conditions show that the IEC IA onshore category has the highest probability of failure, and the IEC IIIA offshore category has the lowest. However, the wind class (I, II, III), whether onshore or offshore, has minimal

impact on the probability of failure, while turbulence intensity (A, B, C) has a significant effect on reliability. The probabilities of failure for the selected onshore and offshore wind sites are generally higher than those of IEC sites.This indicates that IEC-designed turbines may experience increased risk of static overload in blade bearings when applied to certain real wind sites. It reinforces the need for site-specific assessment considering turbulence and manufacturing variability. It also shows the necessity of assessing the blade bearing in every wind site condition according to extreme turbulence wind.

*Code and data availability.*   The code and data used in this study is available upon reasonable request.

*Author contributions.*   AR wrote the original draft. ARN contributed to paper revisions, funding acquisition.

*Competing interests.*   ARN is a member of the editorial board of the Wind Energy Science journal. The authors declare that they have no further conflicts of interest.

*Acknowledgements.*   This research was partially supported by Made4Wind project Under Horizon Europe Research and innovation funding

programme under GA No. 101136096.

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
