# Peer review of "Probability Assessment of Static Overload in Wind Turbine Blade Bearings Considering Turbulence, Design, and Manufacturing Variability"

_Wind Energy Science, 2024_

## Editor Comment (EC1)

Review by editor:

**On reliability design and code calibration of wind turbine blade bearings under extreme wind conditions**

**21 April 2025**

The combination of two highly specialized topics (structural reliability and bearing design) in the present manuscript makes it challenging to find reviewers with such profile – or at least challenging to find reviewers that can cover both topics simultaneously. I therefore see it necessary to supplement the other reviews with a "review by editor", where I am mostly focusing on topics related to the structural reliability and load calculations.

**General comments**

- Grammar: the use of the definite article "the" is not correct in a number of places in the paper, there are both examples of unnecessary "the" and examples of missing "the" or "a". An example with the abstract – the first few sentences should read: "This study presents a reliability analysis of a blade bearing against ultimate limit state failure. The National Renewable Energy Laboratory 5 MW Reference Wind Turbine is selected for the study, and a Monte Carlo simulation (or "the Monte Carlo simulation method") is used for reliability analysis and estimation of the probability of failure....". Please correct the entire manuscript for grammar.
- 2) Blade bearing or blade pitch bearing? In my view, it is better to mention it is a pitch bearing.
- 3) Section 1: The authors list references that deal with fatigue of rolling bearings. However, I am missing the discussion about the fundamentally different loading pattern that blade bearings are subject to they work as so-called "oscillating bearings" which is the main reason for specific failure modes (such as static failure) being more relevant. I suggest the authors discuss the specific loading pattern of oscillating bearings, add any necessary literature, and link the loading pattern with the choice of failure mode.
- 4) Figure 2: I am not sure the uncertainties on the loads side are so clear. I believe they should be split more clearly between aleatory uncertainties (like the short-term wind conditions), and epistemic uncertainties which could be further subdivided into 1) model uncertainties in the climate model, the dynamic loads calculation model and the bearing model, 2) measurement uncertainties, and 3) statistical uncertainties due to finite sampling periods. It is also fine if some of these uncertainties are considered irrelevant or small enough to be omitted, but the current description does not make it clear what is the source of each uncertainty is and why it should be included or not.
- 5) Page 10, line 183: the authors use the maximum likelihood estimator (MLE) for fitting extreme probability distributions. While the MLE is the standard way of fitting parametric probability distributions to data, it is often insufficient when the aim is proper representation of the tails of the data. Further, the current format of Figure 3 is not clearly showing the quality of the fit in the tails. I hereby remind my earlier comment which the authors suggested will be addressed in a revised

version: "When considering reliability analysis with respect to ultimate limit state with small probabilities, the failures normally occur under rare conditions which fall within the tails of the underlying distributions. Therefore, when evaluating the quality of fit of distributions, normally the most useful way of graphical evaluation is plotting the exceedance probabilities (1 minus the CDF) on a logarithmic y-axis. Sometimes also the max likelihood method may not be the most applicable fitting method as it will ensure the best fit to the main body of the distribution but not necessarily the tails. I suggest you replace Figure 3 by an exceedance probability plot, and based on that reevaluate which may be the best fitting distribution, and whether you may need other fitting method than the MLE or other quantitative criteria than the CE indicator. For the exceedance probabilities of the actual data, you can use the empirical CDF formulas based on data ranking."

- 6) Could changes in the bearing geometry affect the maximum ball force?
- 7) Page 14, line 241: Ball diameter: in my view the changes "fine to coarse machining" will not affect the nominal ball diameter, but rather the tolerances which will be larger for coarse machining, correct? If that's the case, then the probability distribution of the ball diameter for "coarse machining" will correspond to a worse-case scenario. Further, we may assume that the diameters for balls within the same bearing will vary, and we will have a population of ball diameters in a single bearing. So, a ball diameter distribution can be taken into account by defining what could be worst-case ball dimension (the ball with highest deviation from the nominal diameter within the bearing) and base the reliability calculation on this worst-case ball dimension. Could the authors discuss how this can affect their assumptions?
- 8) Failure probabilities always are with respect to a certain reference period. What is the reference period here, I believe it is annual failure probability (as in Fig.7)? Please specify, and define  $P_f$  as annual probability of failure the first time you mention it in the text.
- 9) Figure 8, load index: this is a discussion/observation rather than anything that is used further in the paper, right? Maybe extend the discussion on how this information can be used (for example for tuning safety factors).

**Specific comments**

- 10) Abstract: please include a description of the bearing configuration (e.g., a double-row, four-point contact ball bearing).
- 11) Page 1, lines 14-15, "changing the broken blade bearing is costly" I agree, but please support with a reference and/or an indicative number.
- 12) Page 6, equations 2,3,7: I suggest to denote the trigonometric functions with regular text rather than italic, e.g.,  $\sin \alpha$  rather than  $sin\alpha$ .
- 13) Figure 5 d): there is a typo (cotact instead of contact).
- 14) Page 17, line 273: "class IA has a reliability of 0.999979". I suggest changing to "class IA has annual probability of failure of 2.1e-5". In the same sentence, you both define reliability numbers and probability of failure numbers, which is confusing. I suggest to stick with failure probabilities in scientific format.
- 15) Pf: I suggest changing the notation to a formula-like format, such as " $P_f$ ".
- 16) Conclusions: when you summarize the paper in the first paragraph, please add description of the bearing configuration.

---

## Author Comment (AC1)

We would like to thank Dr. Matthias Stammler for his constructive comments, which have improved the quality of our paper. We have addressed all the comments, as presented below.

A very interesting work and a good approach to compare IEC classes with real sites!

This comment collects a few things I found during a first read, with a focus on practical aspects of blade bearings:

Line 13: "Blade bearings serve as the connection point between the rotor and the hub, allowing the blades to rotate around the hub. " In my understanding, the rotor consists of blades, blade bearings, and the hub. Thus I would rather say the blade bearings connect the blades with the hub. An I find it misleading to say the blade rotate around the hub. They rather rotate around their own primary axis?

Answer: The comments are correct, and the sentences are changed as follows:

"Blade bearings serve as the connection point between the blades and the hub, allowing the blades to rotate around their axis."

Line 16: I would recommend to mention cage failures as well.

Answer: The text has been revised as below:

"Blade bearing failure consists of different damage modes, including rolling contact fatigue, core crushing, edge loading, ring fracture, rotational wear, fretting, false brinelling (Andreasen et al., 2022), and cage damage."

Line 20 to 30: There is also a publication by Schwack comparing the different RCF calculation methods.

Answer: The following text is added:

", and Schwack et al. (2016) compared different fatigue lifetime calculation methods and showed huge differences in the calculated lifetime for different approaches."

Line 36 to 42: The plastic deformation of 0.0001D is set a limit value in the ISO76 and the 2009 version of the DG=3. However, the 2024 version of the DG03 opens possibilities to increase this value, mainly for two reasons:

- Pitch bearings in general can operate in raceway conditions that are considered a failure in other applications. Macroscopic spallings are not a reason to stop operating a pitch bearing and will not cause an exchange of it. Only when the risk of inoperability is imminent (i.e. expected friction torque too high for drive or loss of blade connection) an exchange is undertaken.

- In four-point contact ball bearings highest static loads in rolling contacts are at high contact angles. 'Normal' operation in power production is at lower contact angles, thus for the main part of its lift, the ball does not roll over the indent.

I would highly recommend to mention these concepts in your introduction as they heavily influence the general conclusions you possibly draw at the end. Also I would vey much recommend to get acquainted with the concept of damage and failure as described in the 2024 DG03 and use those terms consistently throughout the paper.

Answer: It is correct that in some research, such as the work done by Zwirlein et al. in 1983, with total permanent deformation of 0.0005D, no core crushing occurred. To the authors' knowledge, those studies were mainly based on experience and might be correct for a certain range of bearings with specific material and heat treatment, as stated in the work by Lai et al. 2009. In DG03, 2024 also the value is open, and no specific value is presented. To compare reliability in different wind cases, the ISO 76 assumption is considered, which is a known standard and gives the ability to compare the results. In order to clarify this issue for readers, the following text is added:

"Although some research showed that the slewing bearing can tolerate higher total permanent deformation while no core crushing occurred as stated in DG03, those results might be correct for a certain range of bearings with specific material and heat treatment, as stated in work by Lai. The current work uses the value of 0.0001D; however, there is a possibility that the core crushing damage does not occur in some bearings."

As recommended, the paper was revised according to the concept of damage and failure. However, the term "probability of failure" is kept as a main term in reliability analysis.

Figure 1 Step2: It should be "Blade root loads" instead of "Blade's root loads" I think, because the first is a common expression. Also it is a bit confusing because the sketch shows an airfoil used in the outer portions of the blade, but most certainly not a blade root. I would consider the lift at the blade root to be negligible.

Answer: The comment is correct, and the caption was corrected.

Figure 1 Step3: Are you sure you obtain the load Q in N? Or should it be kN for this graph?

Answer: The load is 104 N. It was presented to show the general distribution of the load inside the bearing. To avoid confusion, the unit has been clarified.

Figure 1 Step4: The sketch does not fit the caption. It shows to spherical bodies in unloaded contact, but certainly not a maximum Hertz stress

Answer: However, the sketch is extracted from DG03, but it does not fit the caption. The figure is changed to a proper one.

Equation 7 please give a reference

Answer: The related reference (equation 6.1 from DG03 (2024)) is added.

Line 108: Capital Z instead of small z

Answer: The text was revised accordingly.

Section 3.2.1 Pitch bearing rings are commonly manufactured of 42CrMo4 steel - please elaborate on the choice of the studies on AISI51200 (100Cr6) steel property distributions - the hardening process is fundamentally different.

Answer: The following text was added to elaborate on the consideration of 42CrMo4 steel.

"Imdad et al. (2024) studied 42CrMo4 steel that was submitted to different heat treatments and hardness levels. The hardness level in different heat treatments has a deviation between 2% and 6.5%."

Section 3.2.2 While it is fundamentally true that all dimensions have a certain variance to them, this is somewhat countered by the assembly process: Rings and balls are matched to obtain a target friction torque in unloaded condition. Thus combining normal distributions for all parameters does not reflect reality.

Answer: Thanks for the valuable information about the assembly process of the slew bearings. In the current study, the intention was to assess the sensitivity of each dimension; therefore, each dimension was studied independently, and combining different normal distributions didn't happen. In order to clarify it, the following text is added to the content:

"It should be noted that each dimension was analyzed independently by the relevant distribution."

Section 4.2.1 There is no such thing as very coarse machining of balls of this size. You usually buy them in batches of very fine tolerances and them match them to obtain target torques.

Answer: The authors acknowledge that one usually buys the balls. We could not find any reference for the tolerances; therefore, we used the machining tolerance for sensitivity analysis. In order to estimate and assess the trend of changes in the reliability due to changes in ball diameter, a broad range of values is considered; however, some values are not realistic. The following text is added to clear this consideration for the reader:

"Although the balls are usually manufactured and sorted in a batch with fine tolerances in diameters, it is assumed that the ball diameter can change from fine to very coarse machining according to ISO 2768-1 ISO 2768-1 (1989)."

"However, the extreme tolerances are not realistic; they can help to observe the trend of changes in reliability. The assumption leads to a 0.15 to 1.5 mm variation in the ball diameter."

Section 4.2.3 Lower values of raceway conformity will drastically increase the friction torque and the likelihood of surface wear. it is questionable to use a value range this big

Answer: It is correct that the lower value of raceway conformity is not so realistic; however, some previous works, such as [Daidie et al., 2008, 3D Simplified Finite Elements Analysis of Load and Contact Angle in a Slewing Ball Bearing] and [Krynke et al., 2012, Modelling the contact between the rolling elements and the raceways of bulky slewing bearings], used or presented a low value for raceway conformity. We tried to expand the raceway conformity to even somewhat unrealistic values to see how much it affects the reliability value and estimate the sensitivity of the reliability to this parameter and its trend.

Section 4.2.4 It would make sense to differentiate between the nominal contact angle (as manufactured) and the 'contact angle' (changes as a function off load and ring deformation in operation). Currently, it is a bit unclear which one the authors refer to.

Answer: The following text is added:

"The initial contact angle in this study referred to the nominal contact angle, which is in a loadfree condition."

---

## Author Comment (AC2)

We would like to thank the editor for helpful comments. Below, we respond to each point and describe the changes made to the manuscript.

The combination of two highly specialized topics (structural reliability and bearing design) in the present manuscript makes it challenging to find reviewers with such profile – or at least challenging to find reviewers that can cover both topics simultaneously. I therefore see it necessary to supplement the other reviews with a "review by editor", where I am mostly focusing on topics related to the structural reliability and load calculations.

**General comments**

1) Grammar: the use of the definite article "the" is not correct in a number of places in the paper, there are both examples of unnecessary "the" and examples of missing "the" or "a". An example with the abstract – the first few sentences should read: "This study presents a reliability analysis of a blade bearing against ultimate limit state failure. The National Renewable Energy Laboratory 5 MW Reference Wind Turbine is selected for the study, and a Monte Carlo simulation (or "the Monte Carlo simulation method") is used for reliability analysis and estimation of the probability of failure...". Please correct the entire manuscript for grammar.

• Answer: A thorough language edit has been performed.

2) Blade bearing or blade pitch bearing? In my view, it is better to mention it is a pitch bearing.

• Answer: Blade bearing and pitch bearing are both common. In order to make it clear for the reader, the following text is added in the introduction: *"with blade bearings, also named pitch bearings,"*

3) Section 1: The authors list references that deal with fatigue of rolling bearings. However, I am missing the discussion about the fundamentally different loading pattern that blade bearings are subject to – they work as so-called "oscillating bearings" which is the main reason for specific failure modes (such as static failure) being more relevant. I suggest the authors discuss the specific loading pattern of oscillating bearings, add any necessary literature, and link the loading pattern with the choice of failure mode.

- Answer: The following text is added in the introduction part:
  - "A common and distinguishing feature of blade bearings is that they involve a rather slow oscillatory motion (Harris et al., 2009). This movement pattern differs from that of bearings in most other industrial applications, where bearings usually rotate continuously (Menck et al., 2020). Unlike most oscillating bearings, blade bearings perform stochastic oscillations rather than constant amplitudes back and forth (Stammler et al., 2024). They oscillate through only a few degrees, up to 20°(Keller and Guo, 2022) during normal power operation, depending on wind conditions and turbine sizes, and up to 90°in emergency feathering, rather than completing full revolutions. The limited rolling distance, long dwell periods at fixed pitch angles, and frequent load reversals with high axial offsets concentrate stress cycles within a small contact zone on the raceways. Blade bearing failure consists of different damage modes, including rolling contact fatigue, core crushing, edge loading, ring fracture, rotational wear,

fretting, false brinelling (Andreasen et al., 2022), and cage damage. Because these modes initiate in the same high-stress regions, the static-overload reliability analysis developed here directly addresses the most critical damage mechanisms for oscillating blade bearings. As a part of the design and certification process in a blade bearing, the static safety factor of the blade bearing in the ultimate limit state (ULS) must be assessed, as mentioned in (IEC 61400-1, 2019; DNV-ST-0437, 2016; Harris et al., 2009; Germanischer Lloyd, 2010; Stammler et al., 2024)."

4) Figure 2: I am not sure the uncertainties on the loads side are so clear. I believe they should be split more clearly between aleatory uncertainties (like the short-term wind conditions), and epistemic uncertainties – which could be further subdivided into 1) model uncertainties in the climate model, the dynamic loads calculation model and the bearing model, 2) measurement uncertainties, and 3) statistical uncertainties due to finite sampling periods. It is also fine if some of these uncertainties are considered irrelevant or small enough to be omitted, but the current description does not make it clear what is the source of each uncertainty is and why it should be included or not.

• Answer: We agree that a clearer distinction between different types of uncertainties would help the reader better understand the structure of the reliability model. Therefore, we have updated Figure 2 to clarify the sources and types of uncertainty involved in the loads. In particular, we now distinguish between:

Aleatory uncertainty represents inherent variability, such as the stochastic nature of short-term wind conditions (e.g., turbulence and seed number variability).

Epistemic uncertainty, which we further divide into:

Model uncertainty — including simplifications in aerodynamic load modeling, and structural response

Measurement and parameter uncertainty

Statistical uncertainty — arising from finite sampling periods.

To make this clearer, we drew a light dotted rectangle around them, labeled "combined  $\rightarrow$  uncertainty on wind-induced loads", and the following text is added right after the figure:

"The dashed box represents uncertainty on loads that have distinct uncertainty sources -(i) external wind variability and (ii) Epistemic uncertainty arises from model, measurement, and stochastic uncertainties, which are sampled separately and then combined to generate the stochastic wind-induced loads. The uncertainties of the measurement and statistics are not considered in this study."

5) Page 10, line 183: the authors use the maximum likelihood estimator (MLE) for fitting extreme probability distributions. While the MLE is the standard way of fitting parametric probability distributions to data, it is often insufficient when the aim is proper representation of the tails of the data. Further, the current format of Figure 3 is not clearly showing the quality of the fit in the tails. I hereby remind my earlier comment which the authors suggested will be addressed in a revised version: "When considering reliability analysis with respect to ultimate limit state with small probabilities, the failures normally occur under rare conditions which fall within the tails of the underlying distributions. Therefore, when evaluating the quality of fit of distributions,

normally the most useful way of graphical evaluation is plotting the exceedance probabilities (1 minus the CDF) on a logarithmic y-axis. Sometimes also the max likelihood method may not be the most applicable fitting method as it will ensure the best fit to the main body of the distribution but not necessarily the tails. I suggest you replace Figure 3 by an exceedance probability plot, and based on that reevaluate which may be the best fitting distribution, and whether you may need other fitting method than the MLE or other quantitative criteria than the CE indicator. For the exceedance probabilities of the actual data, you can use the empirical CDF formulas based on data ranking."

• Answer: Thank you for this valuable comment. We agree that proper representation of the distribution tails is crucial for reliability analysis, especially when evaluating rare events. In our study, the fitted distribution is constructed using extreme load values extracted from each simulation run. Therefore, the body of the resulting distribution is itself composed of extremes, and capturing its overall shape, including the center and tail, is essential.

We chose the Generalized Extreme Value (GEV) distribution because it is widely used for modeling block maxima and rare events. While we applied the maximum likelihood estimator (MLE) for parameter fitting, as is common practice, we acknowledge that MLE may not always provide the best fit to the tail region. To address this, we used the Coefficient of Efficiency (CE) as a goodness-of-fit criterion across the entire distribution, and not just the central tendency. Additionally, the exceedance probability plot has been included as a separate figure.

6) Could changes in the bearing geometry affect the maximum ball force?

• Answer: The analytical expression we use for the maximum ball force is an explicit function of three geometric quantities—pitch-circle diameter Dpw, contact angle  $\alpha$ , and number of balls Z; However, the probability of failure is not sensitive to small variations of these parameters.

7) Page 14, line 241: Ball diameter: in my view the changes "fine to coarse machining" will not affect the nominal ball diameter, but rather the tolerances which will be larger for coarse machining, correct? If that's the case, then the probability distribution of the ball diameter for "coarse machining" will correspond to a worse-case scenario. Further, we may assume that the diameters for balls within the same bearing will vary, and we will have a population of ball diameters in a single bearing. So, a ball diameter distribution can be taken into account by defining what could be worst-case ball dimension (the ball with highest deviation from the nominal diameter within the bearing) and base the reliability calculation on this worst-case ball dimension. Could the authors discuss how this can affect their assumptions?

• Answer: In our study, the balls were assumed to be equal in diameter, and a range of diameters according to ISO 2768 was studied. If we assume a distribution of ball diameters in the bearing, the highest probability of failure won't be exclusively due to ball diameter but also the position of the ball. In other words, it could happen that the ball with the least diameter doesn't experience maximum force, therefore it wouldn't have the highest probability of failure. It should be added that balls are usually

considered in a batch with fine tolerance. The text is revised as follows to clear our intention and assumptions.

"The nominal size of the ball diameter is 75 mm. Although the balls are usually manufactured and sorted in a batch with fine tolerances in diameters, it is assumed that the ball diameter can change from fine to very coarse machining according to ISO 2768-1 (ISO 2768-1, 1989). In every analysis, the balls' diameters are assumed to be the same, and a range of diameters was studied. However, the extreme tolerances are not realistic; they can help to observe the trend of changes in reliability. The assumption leads to a 0.15 to 1.5 mm variation in the ball diameter."

8) Failure probabilities always are with respect to a certain reference period. What is the reference period here, I believe it is annual failure probability (as in Fig.7)? Please specify, and define Pf as annual probability of failure the first time you mention it in the text.

• Answer: Thank you for catching this omission. The reliability calculations are indeed referenced to one year. To make this explicit, we have made the following change: *"The annual probability of failure, Pf,"*

9) Figure 8, load index: this is a discussion/observation rather than anything that is used further in the paper, right? Maybe extend the discussion on how this information can be used (for example for tuning safety factors).

• Answer: You are correct that the load-index plot was originally presented only as an observation. We have now expanded the accompanying text to explain its practical value:

"This observation has important implications for reliability analysis and simulation practice. It shows that using only 15 seeds, as typically done in standard simulations, can lead to a non-negligible underestimation of extreme ball loads, especially in complex wind site conditions. The load index provides a simple and effective measure to quantify this underestimation. It can help practitioners evaluate whether their simulation setup sufficiently captures the load extremes, and when limited seed numbers are used, the load index can offer a basis for applying correction factors to improve the accuracy of failure probability estimates. Additionally, the variation in load index across different wind sites emphasizes the need for site-specific assessment when evaluating blade bearing reliability."

**Specific comments**

10) Abstract: please include a description of the bearing configuration (e.g., a double-row, four-point contact ball bearing).

• Answer: The description of the bearing configuration has been added to the abstract.

11) Page 1, lines 14-15, "changing the broken blade bearing is costly" – I agree, but please support with a reference and/or an indicative number.

• Answer: We have now supported this statement with a relevant reference. As shown in Mishnaevsky and Thomsen (2020), the cost of renting a crane for major repairs such as blade or bearing replacement can reach up to \$350,000 per week. This high logistical cost, primarily related to crane transportation, setup, and operation, is a major contributor to the overall expense of replacing a blade bearing. The manuscript has been updated accordingly.

"The cost of replacement with a crane can reach up to \$350,000 per week (Mishnaevsky Jr and Thomsen, 2020)."

12) Page 6, equations 2,3,7: I suggest to denote the trigonometric functions with regular text rather than italic, e.g.,  $\sin \alpha$  rather than  $sin\alpha$ .

• Answer: The trigonometric functions changed to regular text.

13) Figure 5 d): there is a typo (cotact instead of contact).

• Answer: The caption has been corrected.

14) Page 17, line 273: "class IA has a reliability of 0.999979". I suggest changing to "class IA has annual probability of failure of 2.1e-5". In the same sentence, you both define reliability numbers and probability of failure numbers, which is confusing. I suggest to stick with failure probabilities in scientific format.

• Answer: The sentence has been revised to present the probability of failure and the scientific format only used.

15) Pf: I suggest changing the notation to a formula-like format, such as "Pf".

• Answer: The notation has been changed to formula format (*Pf*) throughout the paper.

16) Conclusions: when you summarize the paper in the first paragraph, please add description of the bearing configuration.

• Answer: The bearing configuration is added in the conclusion as below:

"The presented work studies the static overload probability of failure in the double-row, four-point contact ball blade bearing at the ultimate limit state."

---

## Author Comment (AC3)

We would like to thank the reviewer for the valuable feedback. Below, we respond to each comment and explain the changes made to the manuscript.

In this paper, the authors present a methodology for calculation of the probability of failure of a wind turbine pitch bearing due to static overload – specifically based on exceedance of the static safety factor as determine by the recently published pitch bearing design guide (Stammler et al 2024). The presented methodology also includes treatment of a variety of design uncertainties, which is especially interesting, and examines specific wind sites. Having said that, I have a number of technical and editorial comments. My most significant question pertains to the interrelationship of terms R and S as described in the comments. Maybe there is no issue, but I seek clarification and comment from the authors.

Title

- The phrase "reliability design" is applied very broadly in the title and "reliability analysis" similarly in the text. This is a little misleading, as what has been examined is the risk of static overload resulting in plastic deformation damage, which is just one aspect of the design. Obviously, rolling contact fatigue and wear prevention are other essential aspects in the design.
- Answer: The paper has employed the classical "structural reliability" approach to estimate the reliability of the design under a certain failure mode static overload in this case. We therefore used this term "reliability design" to emphasize the level of reliability one can expect under different design assumptions. We have now updated the text to make this to be more clear.
- Other than in the Title, "code calibration" is not used in the manuscript. What did the authors intend with this phrase? I have the feeling it has something to do with comparison of IEC wind classes to actual wind sites, but I would not describe this as "code calibration". Please reconsider your meaning.
- Answer: In reliability engineering, "code calibration" often refers to checking whether the partial-safety factors or load factors in a design code achieve a target reliability when confronted with real stochastic input data. Our study compares IEC 61400-1 wind classes against measured site-specific wind regimes and quantifies the resulting probability of failure—an exercise that resembles a pre-calibration check of the code's implicit safety margin. We agree that we do not perform a formal statistical calibration of IEC partial factors, nor do we propose new calibrated factors. To avoid over-promising, we have deleted the term and replaced it with "code–site comparison." In sections 4.3-4.4.
- The title changed to the following:

"On reliability assessment of wind turbine blade bearings under extreme wind conditions"

**Abstract**

- Lines 1-5: The Abstract says the manuscript presents and describes "*the* reliability analysis" in several places. As commented on the Title, this manuscript examines the risk of static overload resulting in plastic deformation damage. This is one or "a" aspect of the design, not "the" aspect. Additionally, the abstract mentions "probability of failure" very broadly without specifying that it is limited to risk of static overload resulting in plastic deformation damage. In terms of the importance of ball diameter and the conclusion on IEC vs. actual wind sites, please see later comments.
- Answer: We agree that the opening sentence should state the specific failure mechanism that is analyzed. We now introduce the study as an "*analysis under static overload*".
- Lines 3-6: The phrase "sensitivity in the dimension aspect of reliability" is unclear. That is, it is not clear if "dimension" refers to a physical dimension (ball diameter, pitch diameter) assessed in the reliability analysis, or if it refers to one element or an aspect of the analysis. Further, lines 3-6 all say similar things, but in slightly different manners so, it is not clear if key differences are being communicated or not. My sense is not, so for greater clarity, I recommend lines 3-6 be combined and simplified to something like "The sensitivity of the probability of failure to uncertainties in turbulence intensity, material properties, and bearing dimensions is evaluated. Within the bounds examined, the pitch bearing conformity and ball diameter have the largest effect on the probability of failure."
- Answer: We have adopted the reviewer's suggested wording in full. The text has been revised as below:

"The sensitivity of the probability of failure to uncertainties in turbulence intensity, material properties, and bearing dimensions is evaluated. Within the bounds examined, the blade bearing conformity has the largest effect on the probability of failure, and ball diameter is next."

- Lines 6-7: Here too the sentence is a bit hard to understand, especially "IEC standards...are studied...". I recommend more simply and directly "The probability of failure for some example wind sites around the world is assessed and is higher at those sites than for wind conditions described by IEC 61400-1."
- Answer: The sentence has been shortened and clarified as recommended. The text has been revised as below:

"The probability of damage for case-study wind sites around the world is assessed, and it is observed that the probability of failure is higher in some cases than for wind conditions described by IEC 61400-1."

1 Introduction

• Lines 14-15: I believe a more accurate representation of the cost in Stehly et al 2023 is that "Although the entire pitch system assembly costs less than one percent of the wind turbine Stehly et al. (2023), changing a blade bearing is costly due to the need for lowering of the blade with a large crane." That is, Stehly lists the full system assembly cost (for all 3 blades, including the bearings, motors, controls, batteries, etc.) rather

than only the bearing and I recommend emphasizing that an individual bearing is vanishingly cheap but can have a costly failure ramification.

• Answer: Thank you for the clarification. We have revised the sentence to state explicitly that the < 1 % figure applies to the *whole* pitch-system assembly, while replacing one failed bearing is costly because it requires blade removal with a large crane and results in significant downtime. The text has been revised as below:

"Although the entire pitch system assembly costs less than one percent of the wind turbine (Stehly et al., 2023), changing a blade bearing is costly due to the need to lower the blade with a large crane."

- Lines 17-18: This sentence, especially "perform the calculation of the ultimate limit state" is a bit garbled and incomplete. I believe a better statement of its intent is "As part of the design and certification process, the blade bearing static safety factor must be assessed in the ultimate limit state (ULS) (IEC 61400-1 2019; DNV-ST-0437 2016; Harris et al. 2009; Germanischer Lloyd 2010; Stammler et al. 2024)."
- Answer: We agree, and the text has been revised as below:

"As a part of the design and certification process in a blade bearing, the static safety factor of the blade bearing in the ultimate limit state (ULS) must be assessed, as mentioned in IEC 61400-1 (2019); DNV-ST-0437 (2016); Harris et al. (2009); Germanischer Lloyd (2010); Stammler et al. (2024)."

- Line 20: I recommend deleting the sentence "There are numerous studies on the fatigue of the bearings; however, the studies about the blade bearing are not many." It provides no information, is entirely subjective, and as time moves on, is less and less accurate. My personal opinion is that it is not accurate even today, as I have a library of 6 dozen technical papers and journal articles regarding blade bearings.
- Answer: We agree. The sentence has been deleted, and the text has been revised as below:

"Several studies have analyzed blade bearings. Among them"

- Line 26-27: Although I myself was a co-author on Rezai et al (2023) and the paper does speak to "seed number", the phrase "...shows the importance of seed number in the turbulence wind model at bearing's life" is not the best description of the work and likely to confuse many readers. I believe a better description of Rezai et al (2023) is that it "...assessed the variation in blade bearing fatigue with shear power law exponent, turbulence intensity, and even resulting from each individual turbulent wind time series."
- Answer: Thank you for the suggestion. We have adopted the reviewer's proposed wording, which more accurately summarizes Rezaei et al. (2023). The text has been revised as below:

"Rezaei et al. (2023) studied the blade bearing of the 5 MW NREL reference wind turbine and assessed the variation in blade bearing fatigue with shear power law exponent, turbulence intensity, and even resulting from each individual turbulent wind time series."

- Line 32-33: I recommend deleting the phrase "...blade bearing reliability is not studied thoroughly...". Similar to line 20, this phrase provides no information and is entirely subjective. I also recommend the phrase "...it is not clear what level of reliability one can obtain with the current design process" also be revised. Although true to some extent, a more informative statement would be to refer to the statistics in Haus, Sheng, and Pulikollu (2024) at https://app.box.com/s/ktjzjdxn77omu1cjoy9znymbrynlsw0d. The statistics therein from 55+ GW of wind plant data show that pitch bearings installed pre-2016 perform fairly well, only reaching a 10% replacement rate in 15 years. However, pitch bearings installed post-2016 on larger wind turbines are projected to have a 10% replacement rate in only 7.5 years.
- Answer: We have removed the subjective clause and replaced the following sentence with an evidence-based statement. The text has been revised as below:

"Haus et al. from 55+ GW of wind plant data show that blade bearings installed pre-2016 perform fairly well, only reaching a 10% replacement rate in 15 years. However, blade bearings installed post-2016 on larger wind turbines are projected to have a 10% replacement rate in only 7.5 years."

- Line 34: It isn't clear that ISO 19902 and ISO 19904-1 standards for oil and gas industries are relevant to offshore wind. More broadly, it seems like IEC 61400-3-1 and -3-2 are better references here. Additionally, IEC 61400-8, titled "Design of wind turbine structural components", seems better suited for a general reference than ISO 2394 for general principles on reliability for structures. Do the authors have a particular meaning in mind with the references to oil and gas standards? In what situations would these standards apply to offshore wind?
- Answer: Thank you for the questions. The reliability targets for offshore wind-turbine support structures are still those in the ISO 19900 series; IEC 61400-3-1 (fixed-bottom, 2023) and IEC 61400-3-2 (floating, 2019) adopt those targets by normative reference. IEC 61400-8 (2024) focuses on nacelle- and hub-level structural components and guidance on external conditions. We have therefore rephrased the sentence both in the introduction and the results to cite:

"IEC 61400-1:2019 for on-shore component-level reliability, ISO 19902:2020 and ISO 19904-1:2019 (as invoked by IEC 61400-3-1:2019 / -3-2:2019) for offshore supportstructure reliability, and IEC 61400-8:2024 for nacelle/hub structural design guidance."

• Line 39: I think most readers will find the sentences "ISO 76 (2006) stated that experience shows that a total permanent deformation of 0,0001 of the rolling element diameter at the center of the most heavily loaded rolling element/raceway contact can be tolerated in most bearing applications without the subsequent bearing operation

being impaired. The bearing static failure corresponds to such a permanent deformation" conflicting. That is, "without operation being impaired" and "corresponds to bearing static failure" are conflicting. I recommend changing the second sentence to "In this work, it is proposed that ball-raceway contact stresses approaching the limits corresponding to ISO 76 increase the probability of failure of the bearing."

• Answer: We agree that the wording was contradictory. We have replaced the second sentence with a statement that makes our modelling assumption explicit:

"ISO 76 (2006) stated that experience shows that a total permanent deformation of 0.0001 of the ball diameters at the most heavily loaded contact without the subsequent operation being impaired. In the present study, we treat contact stresses that reach this deformation limit as attaining the ultimate limit state, i.e., stresses approaching the ISO 76 threshold are assumed to represent the onset of static failure and therefore increase the probability of bearing failure."

- Line 42: The phrase "and the formation of cavities in the raceways" was curious to me. Does this refer to the core crushing phenomenon described in Harris et al. 2009 and Stammler et al 2024?
- Answer: The phrase is extracted from [Harris and Kotzalas (2006)], and when the cavities formed in the core of the raceway, it leads to core crushing as it is described in [Harris et al. 2009] and [Stammler et al 2024].

3.2 Safety factor and failure function

- Lines 76-84: Although somewhat relevant, I don't believe the ISO 76 static safety factor S0 = C0a/P0a mentioned here or shown in Step 5 of Figure 1 is used in the remainder of the manuscript. If this is the case, I recommend deleting these lines as not to distract the reader.
- Answer: The method is referenced in IEC 61400-1 section 9.8.4, and that's the reason it is presented in the paper; however, it is not used in the paper. The ISO 76 static-factor paragraph and the associated "Step 5" box in Fig. 1 have been removed. A short bridging sentence now guides the reader directly to the Hertz-stress-based limit-state formulation used in the remainder of the paper.

*"For the reliability model, the static limit state is formulated with the Hertzian contact-stress criterion."*

• Equations 6-8: I am curious how it is handled and might be worth discussing in the manuscript that the contact area parameters a and b in Equation 6 and within the variable R in Equation 8 are dependent on the maximum ball load Qmax in Equation 6 which is the variable S in Equation 8. That is, R = R(S). Is this automatically accounted for in the described methodology? If so, how? This appears to me acknowledged to some extent in line 135.

 Answer: It is correct that a and b are functions of Qmax and R=R(S), and it is accounted for in the calculation. In every Monte-Carlo realization, Qmax is calculated first, and then a and b are evaluated with consideration of the uncertainty in dimensions and Qmax. Finally, R is calculated based on a and b. The following texts are added:

"a and b in the R are functions of the applied maximum ball load Qmax; therefore, R is implicitly a function of S."

"In every Monte-Carlo realization, therefore, Qmax is computed first, then evaluated a, b, and finally R, ensuring that the dependency R(S) is fully captured"

- Line 102: It appears the text here has the opposite sense of Equation 8. Shouldn't the text here say "If the failure function value is less than or equal to the static safety ratio, the bearing is safe; otherwise, the bearing is in a failure state"?
- Answer: When the failure function value is equal to or less than the safety ratio, the bearing is in a failure state. Let's consider value 1 for the safety ratio. When the function value is less than 1, it means that R is smaller than S and the bearing is in a failure condition. In order to clear it for the readers, the text was revised as follows:

"If the failure function value is equal to or smaller than the static safety ratio, the bearing is in a failure state; otherwise, the bearing is in a safe state."

- Lines 107 115: Although I don't disagree, the math here seems a little longer than necessary. I think one can simply take the cube root of Equation 9 and get to Equation 12 quite directly.
- Answer: Replaced the multi-step derivation with:

"Taking the cube root of Equation 6 yields the failure function directly as"

- Line 126: Here R and S are described as resistance and stress. Hearkening back to Equation 6, these are meant to be R = 4200\*pi\*a\*b/1.5 and S = Qmax I believe. I don't disagree that the load S = Qmax partially represents the stress, but so do the terms pi\*a\*b (contact ellipse area) which is part of what is called R (resistance) if I understand correctly. As commented earlier, a and b are dependent on Qmax, that is, R = R(S). I do agree that the stress of 4200 MPa can be thought of as resistance here.
- Answer: The following text is added to the paper:

"These randomnesses can appear in the R representation of the load-capacity term and S the representation of the applied maximum ball load."

• Figure 2: Two boxes here are "Uncertainty on aerodynamic" and "Uncertainty on wind". I'm not entirely sure I understand the distinction. • Answer: We agree that a clearer distinction between different types of uncertainties would help the reader better understand the structure of the reliability model. Therefore, we have updated Figure 2 to clarify the sources and types of uncertainty involved in the loads. In particular, we now distinguish between:

Aleatory uncertainty represents inherent variability, such as the stochastic nature of short-term wind conditions (e.g., turbulence and seed number variability).

Epistemic uncertainty, which we further divide into:

Model uncertainty — including simplifications in aerodynamic load modeling, and structural response

Measurement and parameter uncertainty

Statistical uncertainty — arising from finite sampling periods.

To make this clearer, we drew a light dotted rectangle around them, labelled "combined  $\rightarrow$  uncertainty on wind-induced loads", and the following text is added right after the figure:

"The dashed box represents uncertainty on loads that have distinct uncertainty sources -(i) external wind variability and (ii) Epistemic uncertainty arises from model, measurement, and stochastic uncertainties, which are sampled separately and then combined to generate the stochastic wind-induced loads. The uncertainties of the measurement and statistics are not considered in this study."

**3.2.1 Uncertainty in material**

- In addition to the given citations, I recommend the authors consider adding Lai, J. 2011. "A New Model for the Static Load Rating of Surface-Induction Hardened Bearings." *Evolution* 2:27–32 and Lai, J., P. Ovize, H. Kuijpers, A. Bacchetto, and S. Ioannides. 2009. "Case Depth and Static Capacity of Surface Induction-Hardened Rings." *Journal of ASTM International* 6 (10): 1–16. http://doi.org/10.1520/JAI102630.
- Answer: The following text is added to include Lai works.

"Lai et al. (2009) presented a model for plastic indentation, and they tested it on 42CrMo4 steel. Their model predicted that the contact pressure for causing plastic indentation of  $10^{-4}$ D in the through-hard raceway is 4260 MPa, as well as good validation results. In the extended work, Lai (2011) predicted the contact pressure to be 4270 MPa."

**3.2.3 Uncertainty in loads**

- Lines 174-176: Here again, the focus on "seed number" still feels odd to me, as though this number has a much more important meaning than it really does. It seems much more straightforward to say that "Different realizations of the turbulence produce a Gaussian distribution of TI in the longitudinal wind component due to spatial coherence (Jonkman 2009)" and "Each simulation leads to a time series of distributions... Different simulations result in a series of..." Similarly in lines 193 to 197, different numbers simulations are considered.
- Answer: The text has been revised as follows:

"Different realizations of the turbulence, called "seed number," produce a Gaussian distribution of TI in the longitudinal wind component due to spatial coherence Jonkman (2009)."

**4.2 Sensitivity analysis and Conclusions**

- Lines 237-239: I'm not sure I understand why this discussion of raceway conformity is here compared to Section 4.2.3?
- Answer: The authors intended to emphasize the importance of raceway conformity; however, the comment is correct, and the text moved to section 4.2.3.
- Comparing sections 4.2.1 through 4.2.4 and Figures 5a-d, the discussion here feels like it is missing the major point: That changes in failure probability for groove conformity are 10^3 greater than those for ball diameter, pitch diameter, and contact angle. Isn't this a very important part of the discussion, or am I missing something? That is, the Pf for a groove conformity of 0.545 is like 10-3 and rapidly decreases to 10-5 for 0.525 a similar level for ball diameter, pitch diameter, and contact angle. Or maybe that's the point of the vertical lines that outside this range these conformities aren't realistic? In the Abstract and Conclusions, is it then fair to compare that "Ball diameter and raceway conformity in this aspect have the highest contribution to the reliability of the blade bearing"? From the plots in 5a-d all with different y-scales, it is really hard for a reader to really see this. Why can't all 4 be put on the same plot? It still feels to me that the effect of the groove conformity is far larger effect than the ball diameter, even within the range of vertical lines in Figure 5c.
- Answer: The vertical lines refer to using fine tolerance for ball diameter and calculating the groove conformity as described in section 4.2.3 with lower and higher tolerance in the ball diameter. The effect of groove conformity is dominant, and therefore, the related section in the abstract and conclusion is modified. Another figure is added to show all parameters together.

**4.2.1 Ball diameter**

- In this section, how are the differences in ball diameter applied? Are all balls equal in diameter and a range of diameters studied, or are these differences in diameter present in the bearing for a given simulation? I wonder what inspection methods might be applied by suppliers during assembly I believe it is typical to make an effort to select balls of similar diameter.
- Answer: The balls assumed equal in diameter and a range of diameters according to ISO 2768 were studied. The text has been modified as below to clear the subject:

"Although the balls are usually manufactured and sorted in a batch with fine tolerances in diameters, it is assumed that the ball diameter can change from fine to very coarse machining according to ISO 2768-1 ISO 2768-1 (1989). In every analysis, the balls' diameters are assumed to be the same, and a range of diameters was studied. However, the extreme tolerances are not realistic; they can help to observe the trend of changes in reliability. The assumption leads to a 0.15 to 1.5 mm variation in the ball diameter."

**4.3 IEC wind conditions and 5 Conclusions**

- In this section, I'm not sure a "fair" comparison is being made between results from say 15 seeds to many, many seeds. The design guideline suggests using load factors as described in Section 7.6.2.2 of IEC 61400-1. Although it is buried in the Appendix A of the design guideline, a safety factor of 1.35 and a partial load factor of 1.25 are applied to the average of the highest loads from each of the DLC turbulent seed time series to determine the maximum stress and static safety factor. Can the authors comment on this? That is, these load factors are purposefully applied knowing that only a few simulations aren't enough to represent the maximum loads and thus the maximum stress and risk of exceeding 4200 MPa. Greater importance is placed on this matter in the Conclusions, where it is stated "It is observed that by considering 15 seed numbers, as proposed in the standards and guidelines, the effect of different turbulence conditions cannot be achieved."
- Answer: A safety factor of 1.35 applied to DLC 1.1, which is a normal turbulence model. The studied DLC is 1.3, which is an extreme turbulence model; therefore, only Partial safety factors for loads of 1.35 according to Table 3 of IEC 61400-1 will apply. It should be that the IEC factors are intended to create a single conservative design load when only 15 turbulent seeds are simulated. Our probabilistic study, in contrast, seeks the full failure-probability distribution, so we use many seeds. The text is revised as follows:

"It is observed that by considering 15 seed numbers, as proposed in the standards and guidelines, the distribution of the loads is not represented."

Table 3, Section 4.4 Wind sites and Conclusions

- I am both interested in and curious about the wind site characteristics of the real sites presented in this study compared to IEC classes. I don't think I saw it anywhere: what are Vave and TI for the real sites compared to the IEC classes? Are they appreciably different? If so, how? Is TI much higher? If they're appreciably different, then it should be no surprise that putting a turbine with a pitch bearing designed even for say IEC 1A is a bad idea. Isn't that an important part of the discussion? Without this information, is it really fair to say "The probability of failure for the selected onshore and offshore wind sites are mostly worse than those of IEC sites"?
- Answer: The average wind speed is the same in both wind sites and IEC classes, and it is mentioned in the descriptions of the DLC section. A new table has been added to present the extreme TIs for the real sites. They were different than IEC classes and both higher and lower. The TIs were higher for those two wind sites with a high probability of failure. The conclusion part is changed as below:

"The probabilities of failure for the selected onshore and offshore wind sites are mostly worse than those of IEC sites, which indicates that IEC-designed turbines may result in lower blade bearing reliability, and it shows the necessity of assessing the blade bearing in every wind site condition according to extreme turbulence wind."

**References**

- The citation for Harris, Rumbarger, and Butterfield 2009 leaves Butterfield's name incomplete. That is, it is only "C.P. B.".
- Answer: The incomplete name was corrected
- The doi for Stehly et al 2023 actually takes one to Harris, Rumbarger, and Butterfield 2009. Since I stumbled on this, I also recommend that this citation be updated to the Stehly 2024 edition at doi 10.2172/2479271.
- Answer: The reference doi was corrected and updated.

Minor grammatical comments:

- Line 14: Please ensure consistency in citation style throughout the manuscript, here "Stehly et al. (2023)", in line 17, "(Andreasaen et al., 2022)" (parenthesis w/ comma), and line 42 "[Harris and Kotzalas (2006)]" (square-bracketed w/out comma). Each citation is used in the same manner and thus should have the same style, which in Latex would be \citep for example.
- Answer: Citation was revised to ensure consistency throughout the manuscript.
- Line 27: Should be "...wind turbine and showed the..."
- Answer: The text in the section has been revised.
- Lines 28-29, 169, and 310: I recommend that "fatigue life" be used here instead of just simply "life" (4 places).
- Answer: The text was corrected.
- Line 40: Change "...can cause possibly stress..." to more simply "...can cause stress..." or "...can possibly cause stress...", although "can possibly" is redundant.
- Answer: The text was corrected.
- Line 42: Please add "also" to "can also lead to" to help distinguish the risk of static failure from surface-initiated fatigue failure.
- Answer: The text was corrected.

- Line 43: Single sentences rarely constitute a paragraph. Please combine with the previous paragraph. Additionally, this sentence refers to "main parameter" (singular), when multiple parameters (plural) are examined.
- Answer: The text was corrected.
- Line 85: I don't believe the acronym SF is used elsewhere in the manuscript. If so, please replace with variable S0.
- Answer: The SF changed to S0.
- Line 86 and 90: MPa is used in 86 while megapascals is used in 90. Please define on first use.
- Answer: The text was corrected.
- Line 94: Please italicize parameters a and b in the text.
- Answer: The text was corrected.
- Line 96: The number of balls is listed previously as z, whereas in equation 7 the variable Z is used.
- Answer: The number of balls was changed to "Z" in the whole text.
- Line 98: Variables Dpw, z, and alpha are previously defined in Table 2, so do not need to be defined here.
- Answer: The sentence was removed from the text.
- Line 161: I'm not sure I understand "spherical roller bearings" and "ball diameter". Shouldn't this be roller diameter?
- Answer: The rolling diameter is correct, and the text has changed.
- Line 163: Should be "sensitivity".
- Answer: It was corrected to "sensitivity".

- Line 168: I think simply "turbulence" or "atmospheric turbulence" makes more sense than "turbulence of the wind turbine". I suppose one could say "turbulence acting on the wind turbine" here.
- Answer: the text changed to "turbulence acting on the wind turbine"
- Lines 216 and 236: here this should be "IEC 61400-1".
- Answer: The standard changed to IEC 61400-1.
- Line 239: Should be "0.5%"
- Answer: It changed to "0.5%".
- Line 306: Should be "Pf".
- Answer: It changed to "*Pf*".
- Line 310: "if are used" should be simply "if used".
- Answer: It changed to "*if used*".

---

## Author Comment (AC4)

We thank the reviewer for the thorough and constructive comments. Below, we address each point in detail and indicate the changes made in the revised manuscript.

This article builds upon previous studies on pitch bearing design guide (ISO 76, DG03) and extended the life calculation from a classic deterministic approach to probabilistic one. The latter approach takes into account the uncertainties in wind turbulence, bearing geometry, material hardness level, et al, which can qualitatively rank the most sensitive parameters for its probability of failure

Specific comments are as follows:

1. The life calculation did not root from rolling contact fatigue assumption, instead, it derives from contact stress safety factor. How realistic is to assume structural reliability as the driving failure mode for pitch bearings? Is there evidence from field observations to support this assumption?

Answer: The paper focuses on the static-overload safety factor that limits irreversible local plastic deformation in the raceway and ring. The static rating is derived directly from the maximum Hertzian contact stress. This follows ISO 76 and DG03, which require the designer to demonstrate an adequate margin against permanent deformation before any RCF life is assessed.

DG03 explicitly states that "the damage mode with the most critical consequences is ring cracking of pitch bearings …" and lists other observed modes—cage wear, racewayedge damage, raceway wear, and bolt-connection failures. Ring cracking is lowfrequency but high-consequence (potential blade release). Standards, therefore, make the static check mandatory:

- IEC 61400-1 and DNV-ST-0361 require verification of bearing static capacity in the ultimate-load cases.
- IEC 61400-8 lists a *reliability-based* approach as one of three accepted structural-assessment routes for RNA structure.
- 2. Included in the introduction, the failure modes are mentioned as "rolling contact fatigue, core crushing, edging loading, ring fracture, fretting, false brinelling". Is there any connection of the present study with the top failure modes?

Answer: The present work quantifies the static-overload limit state—local plastic deformation that can progress to ring cracking or core crushing. This choice is not isolated from the other field-observed modes, and also, as stated in the paper and in (Harris and Kotzalas, 2006), this permanent deformation can cause stress concentrations of considerable magnitude and the formation of cavities in the raceways. These indentations, together with conditions of marginal lubrication, can also lead to surface-initiated fatigue damage. In addition, edge-loading fracture is a *localized* expression of the same contact-stress field we compute; when our model predicts high peak Hertzian stress at the raceway edge, that stress map directly indicates heightened edge-loading risk. Thus, by characterizing the static overload response, the study provides the mechanical input that governs—and often initiates—several of the other

dominant failure modes. In order to cover the above clarification, the title, abstract, and introduction have been revised.

3. Degradation criteria G should get close to the failure modes as much as possible. The proposed one based on safety factor only appears an oversimplification. As shown in the results, probability of failure is less than 0.1% for most of the cases. This estimate is much lower than what has been reported in public domain.

Answer: Our study limits the degradation index G to a single trigger: the contact-stress safety factor falling below unity. We did this deliberately because the paper addresses one specific ultimate limit state of static overload. This criterion is one of the assessments in the blade bearing analysis; therefore, the probabilities of failure are not significant. IEC 61400-8 recommended annual probability of failure target is 5x10-4 for wind turbine structural components in ultimate, fatigue, stability, and critical deflection analysis. The introduction is modified to cover the above clarification. In addition, following texts are in the results section:

" (IEC 61400-1, 2019; IEC 61400-8, 2024) set a target value for the nominal failure probability for structural design for extreme and fatigue failure modes for a reference period of one year is 5×10−4 for component class 2. Component class 2 is "safe-life" structural components whose failure may lead to the failure of a major part of a wind turbine, as given in (IEC 61400-8, 2024). All the wind configurations have a lower failure probability than the target value."

"In addition, these two sites have higher failure probabilities than the failure target value for component class 2 aa given in (IEC 61400-8, 2024)."

4. The studied bearing is 3.6m size. Is there any plan to address the structural flexibility as the uncertainty in the study?

Answer: In this study, the effect of structural flexibility, especially from the ring, is not considered. The effect of the flexibility of the ring was previously studied in the works by Menck et al. (2020) and Rezaei et al. (2024). As described in 3.2.4, uncertainty in the maximum ball force, the effect of flexibility is considered by a distribution from recalculation of the work by (Rezaei et. al 2024).

**5. Please correct typos in the paper.**

Answer: The manuscript has been reviewed, and typographical and grammatical errors have been corrected.

---

## Referee Report (RR1)

In this paper, the authors present a methodology for calculation of the probability of static overload of a wind turbine pitch bearing based on exceedance of the static safety factor as determined by the recently published pitch bearing design guide (Stammler et al 2024). Appreciable revisions have been made based on reviewer comments to the initial submission, but my biggest trouble is with understanding the comparison between IEC classes and the 13 real sites. A close second to that is the importance (or not?) of the treatment of uncertainties, which comprise a major portion of the methodology (5 pages or 25% of the manuscript) but almost none of the discussion of results. I offer the following comments for consideration that could still improve the paper.

Title and terminology

- I still believe there are a number of terms that are used loosely and vaguely throughout the title and text, including "reliability assessment", "probability of failure", and "probability of damage" (all 3 are used in the text). I understand the authors' perspective coming from a structural reliability standpoint; however, I still feel that the most accurate description of the work (and a suggested Title that is truly reflective of the work) is "Assessing the probability of static overload of wind turbine blade bearings considering turbulence, design, and manufacturing characteristics". To me, a "reliability assessment" would examine all possible failure modes, including rolling contact fatigue, static overload, core crushing, and wear. "Probability of failure" and "probability of damage" of course are closer to the described work, but a static overload that causes an indent in the bearing is by no means assured to cause a failure of the bearing. If the authors would like to explain in the text that the methodology for this probability assessment is based on structural reliability assessment methodologies, then that is certainly understandable. Finally, I just don't believe that "extreme wind conditions" in the title (but "ultimate limit state" in the Abstract and most of the paper) isn't even all that important compared to considering turbulence, material, and manufacturing characteristics and uncertainties *which are far more novel aspects of the work and comprise the majority of the manuscript*. I respect the author's desire to pick their own title, I'm just being honest that as-is it does not convey why this work matters to those who care about the design, manufacture, and selection of blade bearings. I will say that the Abstract is a much better reflection of the work than the Title – with the caveat that I'm still not entirely sure comparison of IEC wind conditions to the actual sites is really "apples to apples". Please see later comments.

Introduction

- Line 25: I appreciate most of the modifications here, but the new sentence "Because these modes initiate in the same high-stress regions, the static-overload reliability analysis developed here directly addresses the most critical damage mechanisms for oscillating blade bearings" goes a bit too far and probably appears too early in the text. Based on field experience, I would say ring fracture is the most critical failure mode. I do agree it is almost certainly related to the maximum ball load Qmax. Therefore, that portion of the analysis described in the paper is relevant, but the factor R is not well understood for this damage mode compared to the 4,200 MPa for static overload. I believe a sentence more like "Extreme applied loads, ball loads, and bearing material and design parameters are likely related to many pitch bearing failure modes, so the methodology presented here to assess the probability of static overload given their

uncertainties could be tailored to many pitch bearing failure modes." Such a sentence though is better suited later in the Introduction.

- Line 48-56: I think this list of standards is confusing and missing context. They immediately follow a discussion of pitch bearing reliability, so the reader will assume they all directly relate to that subject:
    - I recommend adding what IEC 61400-1 clause 9.8.4 requires of the pitch bearing design, not just generally for components.
    - I recommend adding a short description of the NREL DG03 (Stammler et al 2024) requirements that add to this. I also recommend adding stating in the next few years the NREL DG03 will be replaced with the newly proposed IEC 61400-18.
    - I recommend adding that IEC 61400-8 currently does not explicitly contain pitch bearings in its Scope. That's not to say it might not be valuable – it could be and could be referred to by IEC 61400-18, much like IEC 61400-4 for gearboxes refers to IEC 61400-8 for their structural components.
    - I understand better the references to "offshore support structures" and ISO 19900 series from the authors response; however, I still do not see how a "support structure" (i.e. foundation) pertain to a blade bearing compared to those mentioned above. Without further explanation from the authors how they see that ISO 19900 relates to a pitch bearing, I recommend these be deleted. It is striking that these are mentioned, when standards directly related to the pitch bearing are ignored.
    - "The standards didn't set reliability targets for machinery components" is stated twice in lines 55 and 56. Although this is true, I recommend this be tailored to what the standards do or don't say about the pitch bearing as described above, as that is the subject of the paper.
- Lines 56-69: The first sentence "The current paper studies the reliability of the blade bearing at ULS, with a deeper focus on the effect of the wind" misses much of the content of the manuscript and far understates the novelty of the work. As mentioned earlier, it doesn't fully "study the reliability" as it focuses only the probability of static overload. "The effect of the wind" is a relatively vague expression, compared to how the manuscript treats *uncertainties with wind (i.e. turbulence), load, material, and manufacturing parameters that are far more interesting*. The last sentence "Moreover, a sensitivity analysis on the effect of the bearing's main parameter on the probability of static failure of the blade bearing was performed" is very hard to understand. What is "the main parameter"? Please refer to my comment on "Title and terminology" and line 25. The novelty of this work is in the method (or framework) to assess wind (extreme or otherwise), load, material, and manufacturing uncertainties on the probability of static overload. The method described here could be directly relevant to or applied to other far more interesting failure modes, potentially such as ring cracking, given sufficient understanding of the variable R for each of them. Static overload is just a convenient illustrator for the purposes of the paper, as R is basically = 4,200 MPa. This whole paragraph is really quite important, as it sets the stage for the paper and it simply isn't well constructed currently. I think it goes too far when it attributes all failures to static overload, as generally implied here and specifically stated previously in Line 25.

2.3 Wind sites

- In Table 3, the reference wind speeds and reference turbulence intensities are shown for classes A+, A, B, and C and the 13 actual wind sites are only mentioned. No corresponding characteristics are given for the 13 sites (other than pointing the reader to Rezai and Nejad, 2023). Later, however, the most relevant information for the 13 sites is described in Section 3.3 (titled Description of DLC). There, "extreme" turbulence intensities are listed, but I am not sure I understand what an "extreme" TI is. Is it just the calculated TI at the site as described by IEC 61400-1 (i.e. ratio of the wind speed standard deviation to the mean wind speed), which for most of these are higher than standard IEC classes (and thence "extreme")? I recommend that Table 6 be moved from Section 3.3 to Section 2.3. Or am I entirely missing something? Why is the description of the TI at the sites, which is an important part of the analysis, not in the Section 2.3 Wind sites? Is there something to "extreme" TI, compared to the reference TI? This becomes important later in Section 3.3 where the DLC and turbulence models are introduced. It's even more important in Section 4 and 5 that describe the effect of turbulence on the probability of failure.

**3.1 and Figure 1**

- Line 95: Here it is stated "In the next section, the procedure for each step is presented" when referring to Figure 1. I appreciate the inclusion of a procedural figure; however, I don't see in the paper where Steps 1-4 are discussed, at least not explicitly (i.e. Step 1, Step 2, etc). It is generally understood from Sections 2.1 and 3.3 there is a model of a turbine used to calculate blade loads referred to in Steps 1 and 2. Note that the blade loads in Step 2 (N, L, D, C) are different than the loads in Equation 4 ($F_r$, $F_a$, M). The FEM and MBS models referred to in Step 3 are not used in the procedure, while I honestly can't tell what Step 4 is at all (the figure is not legible), but I don't believe it's represented in the paper. As described in Section 3.2, the blade root loads are used to determine the maximum ball load using Equation 4, and thence the maximum Hertz stress in Equation 2 and the static safety factor in Equation 1. Honestly, this figure misleads the reader. I recommend deleting Figure 1 or significantly revising it to relate to the methodology of the paper itself. A figure showing a pitch bearing, the forces acting on it ($F_r$, $F_a$, M), and relevant bearing dimensions ($D_{pw}$, D, alpha) and contact properties (f, a, b, $Q_{max}$, and $\sigma_{max}$) would be far more relevant here as these are actually discussed in the paper.

**3.2 Safety factor and failure function**

- Line 103: With the recent revisions, the sentence "In this regard, the static safety factor ($S_0$) is the ratio of the allowable ball load to the actual ball load (Harris et al., 2009)" appears to be out of place and contradictory to Equation 1, which immediately follows. I recommend it be deleted.
- Line 150: On the surface, the sentence "Uncertainty in dimension has an effect on the dimensions of the contact area" seems self-evident. However, I believe the intent is that the uncertainties in pitch bearing design dimensions, such as pitch diameter, ball diameter, contact angle, and groove conformity, affect the dimensions of the contact area a and b. Please clarify. This is then further described in Section 3.2.2, so I believe I am correct.

**3.2.3 Uncertainty in loads**

- Line 216: Here $V_r$ must be defined. I believe it means rated wind speed (i.e. 11.4 m/s) from Table 1.

3.3 Description of DLC

- Line 240: I'm not sure I'd say "The DLC 1.3 contributes to an extreme turbulence model (ETM)", I think rather "uses" or "includes" is a more accurate statement.
- Line 242 and Table 6: I am not sure I understand the transition between the discussion of DLC 1.3 and the site characteristics in Table 6 and "extreme" turbulence intensity. As mentioned earlier, why is this Table not in Section 2.3? Are these extreme values calculated differently than normal and thus not comparable to reference TIs? For 10 m/s for instance, the values of 0.65 and 0.87 for 2 of the sites are extremely high. It is no surprise then that later these sites lead to a higher probability of failure compared to a reference turbulence intensity of 0.16 for IEC 1A.

4.1 Probability distribution function

- Section 4.1 and Figures 3 and 4: the text is extremely small. Please recreate the figures with larger text. A good rule of thumb is as large as the main body text in the document. I will also admit that I have trouble following the narrative here, so I'm not entirely sure I understand what is happening. Overall, when I compare the distributions of 15 and 3,000 seeds in Figure 3, they appear relatively similar. Later, 300 seeds are settled on from the trend in Figure and Table 7 (although if I wanted to argue 200 looks fine as well). This appears to be the net conclusion of this section. Overall, it could be simplified I believe. Not being an expert in this area, I really have trouble understanding this section and how this relates to simulations of DLC 1.3 and the resulting annual probability of failure.

4.2 Sensitivity analysis

- This short paragraph simply says "onshore…1A". As mentioned earlier, I believe it would be valuable to restate this was for DLC 1.3 along with some discussion of how frequently these conditions occurred, as the probabilities of failure are given on an annual basis. I believe this relates to Section 4.1, but I'm really not sure.
- Beginning here in Sections 4.2.1 through 4.2.4 and Figure 6, I will admit having difficulty in understanding previous discussion of uncertainties $\chi_d$, $\chi_f$ and $\chi_m$ and the variation in the probability of failure. To be honest, from Figure 6e what I glean is that other than the lowest groove conformities, the *uncertainties* do not have an appreciable effect on Pf. Is this a fair assessment? I wonder what to make of that? Nothing is offered in the text, which really only focuses what happens over the range of mean values (which, I must note, is different than the uncertainties). It is no real surprise that when mean values change, the static capacity, static safety factor, and probability of failure all change similarly. So…is the final conclusion that the treatment of uncertainties that comprised several pages of the manuscript effectively unnecessary? Or am I missing something?

4.2.3 Raceway conformity

- Line 287: The sentence "Consequently, the uncertainty of the raceway conformity with normal distribution with a standard deviation of 0.5% for the uncertainty of dimension, $\chi_d$, is considered" seems to be just a restatement of the analysis parameters, rather than new information. But why are $\chi_f$ and $\chi_m$ not mentioned? Here, I would be interested in discussion of the effect of the uncertainties as they at least have some effect on Pf at low groove conformities.

Figure 6

- This is an important plot, but again the text is very small. Please increase font size.
- Vertical error bars are used to indicate some range or distribution characteristic (maybe standard deviation) of Pf around each mean value. What exactly do the vertical error bars indicate? I don't believe this is stated in the text. Is this the max and min? Or a standard deviation?
- I'm not sure I would call Figure 6e as "combining a to d" as each one is still plotted individually, Figure 6e is a summary of the individual effects.

4.2.4 Contact angle

- Line 293: Please move this description prior to Figure 6.

4.3 IEC wind conditions

- I assume Figure 8 is conducted for the reference parameters of the bearing stated in Table 2, which explains why the Pf for IEC onshore 1A is 2e-5, as this is basically the same as Figure 6. Is this what was done here? Again, additional explanation would be helpful. Then other classes and reference TIs are studied. Having said that, why are no vertical bars shown like in Figure 6? Again, is it because the effect of the uncertainties are negligible? Is this what "The variance of the results is too small, and it indicates that the clusters are closer together, suggesting less diversity and more consistency." It is just no surprise that as TI and average wind speed decrease, the Pf decrease. Knowing how much is valuable. But again my takeaway is that the uncertainties don't matter compared to the mean values.
- Line 314: From the original Title, the authors have added the sentence "This exercise is referred to as a code-site comparison". I will admit I still don't see the value in mentioning this. Here it seems the term "site" refers to something other than the actual wind sites described later in Figure 10. What is being discussed here is the sensitivity of the probability of failure to the number of simulated seeds for different wind classes.

4.4 Wind sites

- It is worth mentioning in this section that the turbulence > 0.6 at Sujawal and > 0.8 at Aysha, compared to IEC 1A which is 0.16. It is true that this is given in Table 6, but other than a number buried in a Table, it is not discussed. The statement is made that these sites "are categorized in the IEC II class while their Pf is higher than the IEC I class wind sites", but nothing is said about their turbulence level. As can be seen from Figure 8, the TI makes a large difference. As far as I can tell from Table 6, these TI are much higher than even class A+. This does not feel like an apples-to-apples comparison.
- Figure 10: Please label the red line at 5e-4 as the acceptable component class 2 "safe-life". It took me several minutes to figure out that's what it was – for a time I thought it was the described maximum Pf of the IEC cases.

5 Conclusions

- Line 342: I recommend "reliability of the blade bearing" be changed to "probability of static overload of the blade bearing".
- Line 341: Here again, only the range of mean value is being discussed, with groove conformity and ball diameter having the greatest impact on Pf. This is true. I am still struck though that the

manuscript spent several pages developing the methodology for treatment of uncertainties in Section 3.2 through 3.2.4 (over 5 pages) and as far as I can tell, they have a negligible effect and no mention of this is made in Section 4 or 5. Then again, maybe I am completely not understanding the meaning of the vertical bars in Figure 6.

Minor grammatical comments:

- Line 10: the citation style here is better as "…Emissions by 2050 (IEA, 2023)." Maybe I'm overly fussing about it, but I recommend the authors review the correct use of \citet{} and \citep{} (if using LaTex) depending on how the remainder of the sentence is written. This occurs in many places in the text.
- Line 21: a space is needed between 90 degrees and in.
- Please italicize the D in "…groove radius/$D$" in Table 2.
- Line 291: there are extra spaces between 25 and 65 and the degree symbol.

---

## Referee Report (RR2)

In this paper, the authors present a methodology for calculation of the probability of static overload of a wind turbine pitch bearing based on exceedance of the static safety factor as determined by the recently published pitch bearing design guide (Stammler et al 2024). Appreciable revisions have been made based on reviewer comments to the revised submission and I believe these revisions significantly improve the paper. I only offer a few very minor and final clarifications. Pending these, I recommend the manuscript be accepted for publication.

1 Introduction

- Line 62: New text here refers to a "clear" stress threshold. Although 4200 MPa is discussed in ISO 76 and the DG03, both acknowledge that 4200 MPa is an approximate value for ball bearings. Indeed, the DG03 and Lai (2009) do refer to the possibility of higher values in some situations. I recommend that "clear" simply be deleted here.
- Line 74: Similar to the previous comment, I believe it would be clearer and better tie to the rest of the manuscript to 4200 MPa here rather than 0.0001D. That is "The current work assumes a contact stress of 4200 MPa as the criteria for static overload; however, there is a possibility that indentation and core crushing damage do not occur in all bearings at this level."

2.3 Wind sites

- Text between Table 3 and Table 4: I am glad the discussion of the wind sites has been moved here; however, it can still be improved. Table 3 includes the typical reference TIs. Table 4 then "leaps" to extreme TIs (to 4 significant digits…is that really necessary?). These extreme TIs are described as the worst of the worst (i.e. maximum value at each wind speed at each site rather than average value). Missing between the two is the TI for the ETM as this is also later used in much of the analysis. *What would be helpful here is some text and a plot similar to that provided in the authors' response that compares the TI for ETM and the extreme TI for at least a few of the sites – especially Aysha and Kebribeyah*. The main point being that these 2 sites have extreme turbulence levels well above that of even ETM for class 1A (or maybe even A+) and certainly above NTM for class 1A. Others are relatively similar. Section 2.3 would then serve as a nice preview of Section 3.3 and 4.

Minor grammatical comments:

- Line 31: Because this is an inline citation, an "and" is needed between Germanischer Lloyd (2010); and Stammler et al. (2024).
- Table 2: I recommend variables Z and I be italicized like other variables.
- Line 95: A space is missing at ".Some".

---

## Author Response (AR2)

We thank the reviewer for the thorough and constructive feedback. Below, we address each point in detail and indicate the corresponding changes in the revised manuscript.

In this paper, the authors present a methodology for calculation of the probability of static overload of a wind turbine pitch bearing based on exceedance of the static safety factor as determined by the recently published pitch bearing design guide (Stammler et al 2024). Appreciable revisions have been made based on reviewer comments to the initial submission, but my biggest trouble is with understanding the comparison between IEC classes and the 13 real sites. A close second to that is the importance (or not?) of the treatment of uncertainties, which comprise a major portion of the methodology (5 pages or 25% of the manuscript) but almost none of the discussion of results. I offer the following comments for consideration that could still improve the paper.

Title and terminology

- I still believe there are a number of terms that are used loosely and vaguely throughout the title and text, including "reliability assessment", "probability of failure", and "probability of damage" (all 3 are used in the text). I understand the authors' perspective coming from a structural reliability standpoint; however, I still feel that the most accurate description of the work (and a suggested Title that is truly reflective of the work) is "Assessing the probability of static overload of wind turbine blade bearings considering turbulence, design, and manufacturing characteristics". To me, a "reliability assessment" would examine all possible failure modes, including rolling contact fatigue, static overload, core crushing, and wear. "Probability of failure" and "probability of damage" of course are closer to the described work, but a static overload that causes an indent in the bearing is by no means assured to cause a failure of the bearing. If the authors would like to explain in the text that the methodology for this probability assessment is based on structural reliability assessment methodologies, then that is certainly understandable. Finally, I just don't believe that "extreme wind conditions" in the title (but "ultimate limit state" in the Abstract and most of the paper) isn't even all that important compared to considering turbulence, material, and manufacturing characteristics and uncertainties *which are far more novel aspects of the work and comprise the majority of the manuscript*. I respect the author's desire to pick their own title, I'm just being honest that as-is it does not convey why this work matters to those who care about the design, manufacture, and selection of blade bearings. I will say that the Abstract is a much better reflection of the work than the Title – with the caveat that I'm still not entirely sure comparison of IEC wind conditions to the actual sites is really "apples to apples". Please see later comments.

  Regarding the use of terms such as "reliability assessment," "probability of failure," and "probability of damage," we agree that these should be used consistently and accurately. In the revised manuscript, we have taken care to clarify that the study focuses on the static overload limit state, and that this represents only one potential damage mechanism in blade bearings. The phrase "reliability assessment" is now explicitly framed as a structural reliability approach applied to this specific limit state, not to bearing reliability as a whole.

  In line with the reviewer's comment, we also revised our use of "probability of failure" and "probability of damage" to reflect a single overload criterion, and we use this terminology more consistently throughout the paper.

As for the title, we have carefully considered the reviewer's suggested alternative. We have revised the title to the following:

*"Probability Assessment of Static Overload in Wind Turbine Blade Bearings Considering Turbulence, Design, and Manufacturing Variability"*

Introduction

- Line 25: I appreciate most of the modifications here, but the new sentence "Because these modes initiate in the same high-stress regions, the static-overload reliability analysis developed here directly addresses the most critical damage mechanisms for oscillating blade bearings" goes a bit too far and probably appears too early in the text. Based on field experience, I would say ring fracture is the most critical failure mode. I do agree it is almost certainly related to the maximum ball load Qmax. Therefore, that portion of the analysis described in the paper is relevant, but the factor R is not well understood for this damage mode compared to the 4,200 MPa for static overload. I believe a sentence more like "Extreme applied loads, ball loads, and bearing material and design parameters are likely related to many pitch bearing failure modes, so the methodology presented here to assess the probability of static overload given their uncertainties could be tailored to many pitch bearing failure modes." Such a sentence, though, is better suited later in the Introduction.

  The section was modified and moved to a later part of the introduction.
  *"Many of these failure mechanisms initiate in high-stress regions associated with maximum ball loads. While this study focuses on static overload, applied loads and bearing parameters that drive this mechanism are also relevant to other failure modes."*
  *"While the analysis is tailored to one specific failure mode, the approach—based on uncertainties in turbulence, geometry, and material strength—could be extended to assess other damage mechanisms that are similarly influenced by maximum loads and bearing design characteristics."*

- Line 48-56: I think this list of standards is confusing and missing context. They immediately follow a discussion of pitch bearing reliability, so the reader will assume they all directly relate to that subject:
  - I recommend adding what IEC 61400-1 clause 9.8.4 requires of the pitch bearing design, not just generally for components.
  - I recommend adding a short description of the NREL DG03 (Stammler et al 2024) requirements that add to this. I also recommend adding stating in the next few years the NREL DG03 will be replaced with the newly proposed IEC 61400-18.
  - I recommend adding that IEC 61400-8 currently does not explicitly contain pitch bearings in its Scope. That's not to say it might not be valuable – it could be and could be referred to by IEC 61400-18, much like IEC 61400-4 for gearboxes refers to IEC 61400-8 for their structural components.
  - I understand better the references to "offshore support structures" and ISO 19900 series from the authors response; however, I still do not see how a "support structure" (i.e. foundation) pertain to a blade bearing compared to those mentioned above. Without further explanation from the authors how they see that ISO 19900 relates to a

pitch bearing, I recommend these be deleted. It is striking that these are mentioned, when standards directly related to the pitch bearing are ignored.

    o  "The standards didn't set reliability targets for machinery components" is stated twice in lines 55 and 56. Although this is true, I recommend this be tailored to what the standards do or don't say about the pitch bearing as described above, as that is the subject of the paper.

The text was replaced by the following text:

*"The design of blade bearings is governed by a combination of turbine-level and component-level standards. IEC 61400-1, Clause 9.8.4, requires that blade bearings demonstrate a minimum static safety factor against permanent deformation in the ultimate load cases. This requirement is based on limiting the local contact stress between the balls and raceways to a threshold value. DG03 expands on this by recommending a contact stress limit of 4200 MPa, based on ISO 76, and defining a methodology to evaluate loads, contact geometry, and bearing strength. DG03 is widely used in the industry but is expected to be superseded in the coming years by the proposed IEC 61400-18, which will standardize pitch and yaw bearing design. While (IEC 61400-8, 2024) provides structural design guidance for nacelle and hub components, it does not explicitly include blade bearings within its scope. Nonetheless, it may become a valuable reference if adopted in future standards such as IEC 61400-18, just as (IEC 61400-4, 2025) references it for gearbox structural components."*

- Lines 56-69: The first sentence "The current paper studies the reliability of the blade bearing at ULS, with a deeper focus on the effect of the wind" misses much of the content of the manuscript and far understates the novelty of the work. As mentioned earlier, it doesn't fully "study the reliability" as it focuses only the probability of static overload. "The effect of the wind" is a relatively vague expression, compared to how the manuscript treats uncertainties with wind (i.e. turbulence), load, material, and manufacturing parameters that are far more interesting. The last sentence "Moreover, a sensitivity analysis on the effect of the bearing's main parameter on the probability of static failure of the blade bearing was performed" is very hard to understand. What is "the main parameter"? Please refer to my comment on "Title and terminology" and line 25. The novelty of this work is in the method (or framework) to assess wind (extreme or otherwise), load, material, and manufacturing uncertainties on the probability of static overload. The method described here could be directly relevant to or applied to other far more interesting failure modes, potentially such as ring cracking, given sufficient understanding of the variable R for each of them. Static overload is just a convenient illustrator for the purposes of the paper, as R is basically = 4,200 MPa. This whole paragraph is really quite important, as it sets the stage for the paper and it simply isn't well constructed currently. I think it goes too far when it attributes all failures to static overload, as generally implied here and specifically stated previously in Line 25.
The text replaced by the following text:
*"This paper analyzes the probability of static overload in a blade bearing at the ultimate limit state (ULS) using a structural reliability framework, with a deeper focus on the effect of the wind conditions—particularly turbulence intensity—as well as uncertainties in bearing loads, material strength, and manufacturing tolerances. The goal is to quantify the probability that*

*static contact stress exceeds a specified limit, not to assess all failure modes. While static overload is used here as a representative limit state due to its clear stress threshold (4200 MPa), the methodology is generalizable. Extreme loading, combined with uncertainty in geometry and material properties, is relevant to other critical failure modes—such as ring cracking—if the appropriate limit-state definition is available. A sensitivity analysis is also performed to identify which parameters (e.g., raceway conformity, ball diameter) most strongly influence the probability of static overload."*

2.3 Wind sites

- In Table 3, the reference wind speeds and reference turbulence intensities are shown for classes A+, A, B, and C and the 13 actual wind sites are only mentioned. No corresponding characteristics are given for the 13 sites (other than pointing the reader to Rezai and Nejad, 2023). Later, however, the most relevant information for the 13 sites is described in Section 3.3 (titled Description of DLC). There, "extreme" turbulence intensities are listed, but I am not sure I understand what an "extreme" TI is. Is it just the calculated TI at the site as described by IEC 61400-1 (i.e. ratio of the wind speed standard deviation to the mean wind speed), which for most of these are higher than standard IEC classes (and thence "extreme")? I recommend that Table 6 be moved from Section 3.3 to Section 2.3. Or am I entirely missing something? Why is the description of the TI at the sites, which is an important part of the analysis, not in the Section 2.3 Wind sites? Is there something to "extreme" TI, compared to the reference TI? This becomes important later in Section 3.3 where the DLC and turbulence models are introduced. It's even more important in Section 4 and 5 that describe the effect of turbulence on the probability of failure.
The extreme turbulence value table is moved to section 2.3.
IEC 61400-1 defines two different turbulence models: Normal and Extreme. In IEC 61400, the normal turbulence model (NTM) follows equation 10, but the extreme turbulence model (ETM) follows equation 20. ETM is higher than NTM because it represents extreme and rare cases and therefore exceeds NTM across all wind speeds. The reference TI commonly quoted for turbulence classes is defined at Vhub = 15 m/s under NTM, and it differs from extreme turbulence. The plot of ETM and NTM for IEC class IA is shown below.

[Figure]

In this study, extreme turbulence intensity means the maximum turbulence intensity in each wind speed range at the site, and extreme turbulence intensity for the IEC wind classes is directly calculated by TurbSim.

3.1 and Figure 1

- Line 95: Here it is stated "In the next section, the procedure for each step is presented" when referring to Figure 1. I appreciate the inclusion of a procedural figure; however, I don't see in the paper where Steps 1-4 are discussed, at least not explicitly (i.e. Step 1, Step 2, etc). It is generally understood from Sections 2.1 and 3.3 there is a model of a turbine used to calculate blade loads referred to in Steps 1 and 2. Note that the blade loads in Step 2 (N, L, D, C) are different than the loads in Equation 4 (Fr, Fa, M). The FEM and MBS models referred to in Step 3 are not used in the procedure, while I honestly can't tell what Step 4 is at all (the figure is not legible), but I don't believe it's represented in the paper. As described in Section 3.2, the blade root loads are used to determine the maximum ball load using Equation 4, and thence the maximum Hertz stress in Equation 2 and the static safety factor in Equation 1. Honestly, this figure misleads the reader. I recommend deleting Figure 1 or significantly revising it to relate to the methodology of the paper itself. A figure showing a pitch bearing, the forces acting on it (Fr, Fa, M), and relevant bearing dimensions (Dpw, D, alpha) and contact properties (f, a, b, Qmax, and sigmamax) would be far more relevant here as these are actually discussed in the paper.
  The figure has been revised to present the forces on the blade bearing, relevant bearing dimensions, contact properties, and static safety factors. A description of each procedure is presented after the figure.

3.2 Safety factor and failure function

- Line 103: With the recent revisions, the sentence "In this regard, the static safety factor (S0) is the ratio of the allowable ball load to the actual ball load (Harris et al., 2009)" appears to be out

of place and contradictory to Equation 1, which immediately follows. I recommend it be deleted.

The text was removed to avoid redundancy and potential contradiction.

- Line 150: On the surface, the sentence "Uncertainty in dimension has an effect on the dimensions of the contact area" seems self-evident. However, I believe the intent is that the uncertainties in pitch bearing design dimensions, such as pitch diameter, ball diameter, contact angle, and groove conformity, affect the dimensions of the contact area a and b. Please clarify. This is then further described in Section 3.2.2, so I believe I am correct.

  The statement is correct, and here the dimensions refer to bearing design parameters. The text has been revised as follows to make it clear.

  *"The uncertainties in blade bearing design dimensions, such as pitch diameter, ball diameter, contact angle, and groove conformity, affect the dimensions of the contact area a and b"*

**3.2.3 Uncertainty in loads**

- Line 216: Here Vr must be defined. I believe it means rated wind speed (i.e. 11.4 m/s) from Table 1.

  $V_r$ denotes rated wind speed (i.e. 11.4 m/s). The definition of $V_r$ is added to the text.

**3.3 Description of DLC**

- Line 240: I'm not sure I'd say "The DLC 1.3 contributes to an extreme turbulence model (ETM)", I think rather "uses" or "includes" is a more accurate statement.

  The phrase is corrected accordingly.

- Line 242 and Table 6: I am not sure I understand the transition between the discussion of DLC 1.3 and the site characteristics in Table 6 and "extreme" turbulence intensity. As mentioned earlier, why is this Table not in Section 2.3? Are these extreme values calculated differently than normal and thus not comparable to reference TIs? For 10 m/s for instance, the values of 0.65 and 0.87 for 2 of the sites are extremely high. It is no surprise then that later these sites lead to a higher probability of failure compared to a reference turbulence intensity of 0.16 for IEC 1A.

  The table moved to section 2.3. As stated previously, the calculation of turbulence intensity is the same as normal turbulence intensity, but these values are extreme cases of turbulence intensities of one year of wind data. As mentioned before reference turbulence intensity referred to the wind speed of 15 m/s in the normal turbulence wind model. For example, as shown in the figure related to the comment on Section 2.3, normal turbulence of wind speed 15 m/s in IEC 1A is equivalent to 0.18, while its extreme turbulence is 0.26. For wind speed 10 m/s, TI in NTM and ETM are 0.21 and 0.34, respectively. The values of 0.65 and 0.87 in the sites are extremely high, and we accept that it is the main reason for the high probability of failure. These

values show the importance of considering the real wind site condition instead of IEC standard value for ETM.

**4.1 Probability distribution function**

- Section 4.1 and Figures 3 and 4: the text is extremely small. Please recreate the figures with larger text. A good rule of thumb is as large as the main body text in the document. I will also admit that I have trouble following the narrative here, so I'm not entirely sure I understand what is happening. Overall, when I compare the distributions of 15 and 3,000 seeds in Figure 3, they appear relatively similar. Later, 300 seeds are settled on from the trend in Figure and Table 7 (although if I wanted to argue 200 looks fine as well). This appears to be the net conclusion of this section. Overall, it could be simplified I believe. Not being an expert in this area, I really have trouble understanding this section and how this relates to simulations of DLC 1.3 and the resulting annual probability of failure.

    The figure was recreated with a larger text. Different seed numbers were studied in order to obtain a suitable seed number. Each of these seed number sets represents a probability distribution. By increasing the seed number, a bigger number of realizations is created, and the accuracy of the result is higher, but on the other hand, the simulation time will increase. In order to quantify the process of seed selection, it is important to quantify the goodness of fit of the distributions and the convergence of the results. This section is responsible for quantifying the selection of a suitable seed number. A text was added to this section to clear our intention.

**4.2 Sensitivity analysis**

- This short paragraph simply says "onshore...1A". As mentioned earlier, I believe it would be valuable to restate this was for DLC 1.3 along with some discussion of how frequently these conditions occurred, as the probabilities of failure are given on an annual basis. I believe this relates to Section 4.1, but I'm really not sure.

    *The paragraph is revised.*

    *"The probability of failure in the bearing with variation in the ball diameter, pitch circle diameter, conformity, and contact angle is studied. The onshore wind field with an extreme turbulence intensity grade of IA according to IEC 61400-1, DLC 1.3, is considered. $10^8$ samples were considered in the simulation with the Monte Carlo method, and this process was repeated 20 times."*

    DLC 1.3 is an extreme and ultimate load case, and it does not happen frequently. In our study, the values of turbulence intensity of the wind sites are the maximum for one year; consequently, they could happen once a year.

- Beginning here in Sections 4.2.1 through 4.2.4 and Figure 6, I will admit having difficulty in understanding previous discussion of uncertainties χd, χf and χm and the variation in the probability of failure. To be honest, from Figure 6e what I glean is that other than the lowest groove conformities, the *uncertainties* do not have an appreciable effect on Pf. Is this a fair assessment? I wonder what to make of that? Nothing is offered in the text, which really only

focuses what happens over the range of mean values (which, I must note, is different than the uncertainties). It is no real surprise that when mean values change, the static capacity, static safety factor, and probability of failure all change similarly. So...is the final conclusion that the treatment of uncertainties that comprised several pages of the manuscript effectively unnecessary? Or am I missing something?

$\chi d$, $\chi f$, and $\chi m$, are the distributions that are used for the variation of each of the dimensions, force, and material. Indeed, uncertainty in the three main parameters of ball diameter, pitch circle diameter, and contact angle does not have a significant effect on pf, ball diameter has an effect, but not significantly, while the reliability of the bearing can change from $10^{-8}$ to $10^{-2}$ within 5.5% changes in the groove conformity. This effect is important as it shows that with small changes in this value, the probability of failure increases or decreases. This phenomenon is considered for the calculation of the probability of failure in all the other calculations for the IEC case and wind site, as stated in Section 4.2.3.

4.2.3 Raceway conformity

- Line 287: The sentence "Consequently, the uncertainty of the raceway conformity with normal distribution with a standard deviation of 0.5% for the uncertainty of dimension, $\chi d$, is considered" seems to be just a restatement of the analysis parameters, rather than new information. But why are $\chi f$ and $\chi m$ not mentioned? Here, I would be interested in discussion of the effect of the uncertainties as they at least have some effect on Pf at low groove conformities.

  In line 287, it is stated that only groove conformity is assumed in further simulation as an uncertainty of dimension because it has a dominant effect. $\chi f$ and $\chi m$ are defined in Table 6 and stay the same throughout the whole paper; therefore, it is not mentioned again. Otherwise, they should state in every section what is redundant to our understanding. The following text is added to show which variables are considered as an uncertainty in dimension.

  "*The initial contact angle, pitch circle diameter, and ball diameter do not have a significant effect on failure probability and are not considered in the paper as an uncertainty variable regarding dimension*."

Figure 6

- This is an important plot, but again the text is very small. Please increase font size.
  The font size is modified.

- Vertical error bars are used to indicate some range or distribution characteristic (maybe standard deviation) of Pf around each mean value. What exactly do the vertical error bars indicate? I don't believe this is stated in the text. Is this the max and min? Or a standard deviation?
  The error bars are related to the max and min values. The definition of the error bars is added to the text.

- I'm not sure I would call Figure 6e as "combining a to d" as each one is still plotted individually, Figure 6e is a summary of the individual effects.
  The caption is corrected.

**4.2.4 Contact angle**

- Line 293: Please move this description prior to Figure 6.
  The description moved before Figure 6.

**4.3 IEC wind conditions**

- I assume Figure 8 is conducted for the reference parameters of the bearing stated in Table 2, which explains why the Pf for IEC onshore 1A is 2e-5, as this is basically the same as Figure 6. Is this what was done here? Again, additional explanation would be helpful. Then other classes and reference TIs are studied. Having said that, why are no vertical bars shown like in Figure 6? Again, is it because the effect of the uncertainties are negligible? Is this what "The variance of the results is too small, and it indicates that the clusters are closer together, suggesting less diversity and more consistency." It is just no surprise that as TI and average wind speed decrease, the Pf decrease. Knowing how much is valuable. But again my takeaway is that the uncertainties don't matter compared to the mean values.
  Figure 8 is conducted for the reference parameter of Table 2 for onshore IA. An additional explanation is added.
  There is a difference between cluster and uncertainty. Uncertainty is considered in each simulation by applying the probability distribution in force, dimension, and material. In order to see the consistency and repeatability of the simulation, it is repeated 20 times for each IEC wind class and wind sites to make a cluster. Those vertical lines were the variation between different simulations and not uncertainties.

- Line 314: From the original Title, the authors have added the sentence "This exercise is referred to as a code-site comparison". I will admit I still don't see the value in mentioning this. Here it seems the term "site" refers to something other than the actual wind sites described later in Figure 10. What is being discussed here is the sensitivity of the probability of failure to the number of simulated seeds for different wind classes.
  The sentence was removed.

**4.4 Wind sites**

- It is worth mentioning in this section that the turbulence > 0.6 at Sujawal and > 0.8 at Aysha, compared to IEC 1A which is 0.16. It is true that this is given in Table 6, but other than a number buried in a Table, it is not discussed. The statement is made that these sites "are categorized in the IEC II class while their Pf is higher than the IEC I class wind sites", but nothing is said about their turbulence level. As can be seen from Figure 8, the TI makes a large difference. As far as I

can tell from Table 6, these TI are much higher than even class A+. This does not feel like an apples-to-apples comparison.

As mentioned before, the IEC turbulence intensity in ETM is not 0.16; however, to distinguish the difference between these two sites and IECs, the following text is assessed.

*"These high Pf are the result of high turbulence intensity, as addressed in Table 4."*

- Figure 10: Please label the red line at 5e-4 as the acceptable component class 2 "safe-life". It took me several minutes to figure out that's what it was – for a time I thought it was the described maximum Pf of the IEC cases.

  The figure is modified to clarify what the red line is.

**5 Conclusions**

- Line 342: I recommend "reliability of the blade bearing" be changed to "probability of static overload of the blade bearing".

  Text is corrected accordingly.

- Line 341: Here again, only the range of mean value is being discussed, with groove conformity and ball diameter having the greatest impact on Pf. This is true. I am still struck though that the manuscript spent several pages developing the methodology for treatment of uncertainties in Section 3.2 through 3.2.4 (over 5 pages) and as far as I can tell, they have a negligible effect and no mention of this is made in Section 4 or 5. Then again, maybe I am completely not understanding the meaning of the vertical bars in Figure 6.

  As discussed before, the vertical line is related to variation in the cluster and not uncertainty itself. It shows how much the simulation is repeatable and does not have any relation to uncertainty.

Minor grammatical comments:

- Line 10: the citation style here is better as "...Emissions by 2050 (IEA, 2023)." Maybe I'm overly fussing about it, but I recommend the authors review the correct use of \citet{} and \citep{} (if using LaTex) depending on how the remainder of the sentence is written. This occurs in many places in the text.

  The citation is reviewed and corrected throughout the whole paper.

- Line 21: a space is needed between 90 degrees and in.

  The text is corrected.

- Please italicize the D in "...groove radius/$D$" in Table 2.

  The text is corrected.

- Line 291: there are extra spaces between 25 and 65 and the degree symbol.

  The text is corrected.

---

## Author Response (AR3)

We thank the reviewer for the thorough and constructive feedback. Below, we address each point in detail and indicate the corresponding changes in the revised manuscript.

In this paper, the authors present a methodology for calculation of the probability of static overload of a wind turbine pitch bearing based on exceedance of the static safety factor as determined by the recently published pitch bearing design guide (Stammler et al 2024). Appreciable revisions have been made based on reviewer comments to the revised submission and I believe these revisions significantly improve the paper. I only offer a few very minor and final clarifications. Pending these, I recommend the manuscript be accepted for publication.

1 Introduction

- Line 62: New text here refers to a "clear" stress threshold. Although 4200 MPa is discussed in ISO 76 and the DG03, both acknowledge that 4200 MPa is an approximate value for ball bearings. Indeed, the DG03 and Lai (2009) do refer to the possibility of higher values in some situations. I recommend that "clear" simply be deleted here.
  The word "clear" is removed.

- Line 74: Similar to the previous comment, I believe it would be clearer and better tie to the rest of the manuscript to 4200 MPa here rather than 0.0001D. That is "The current work assumes a contact stress of 4200 MPa as the criteria for static overload; however, there is a possibility that indentation and core crushing damage do not occur in all bearings at this level."
  The text is modified as below
  *"The current work assumes a contact stress of 4200 MPa as the criterion for static overload; however, there is a possibility that indentation and core crushing damage do not occur in all bearings at this level."*

2.3 Wind sites

- Text between Table 3 and Table 4: I am glad the discussion of the wind sites has been moved here; however, it can still be improved. Table 3 includes the typical reference TIs. Table 4 then "leaps" to extreme TIs (to 4 significant digits…is that really necessary?). These extreme TIs are described as the worst of the worst (i.e. maximum value at each wind speed at each site rather than average value). Missing between the two is the TI for the ETM as this is also later used in much of the analysis. *What would be helpful here is some text and a plot similar to that provided in the authors' response that compares the TI for ETM and the extreme TI for at least a few of the sites – especially Aysha and Kebribeyah*. The main point being that these 2 sites have extreme turbulence levels well above that of even ETM for class 1A (or maybe even A+) and certainly above NTM for class 1A. Others are relatively similar. Section 2.3 would then serve as a nice preview of Section 3.3 and 4.
  The following text is added to this section.
  *"In this study, the extreme turbulence model (ETM) was investigated. ETM is calculated according to IEC 61400-1 (2019) and prescribes rarer, higher-turbulence realizations that are*

*used later in our ULS analysis. The extreme turbulence intensity at wind sites refers to the maximum turbulence intensity within each wind speed range. To connect the IEC references to the site data, the ETM turbulence intensity curve for IEC classes was computed and compared with the per-bin extreme TI observed at representative sites. Figure 1 shows that Aysha and Kebribeyah exhibit extreme TI values substantially above ETM for Class IA (and, at some wind speeds, even above IA+), while Thanh Hai, Mil Nader, and Flatirons are closer to ETM."*

In addition, a plot is added to the text to clarify the comparison between the wind sites and IEC ETM.

Regarding 4 significant digits in TI, it should be noted that software like TurbSim accepts turbulence intensity in percent; therefore, we present TI with 4 digits.

Minor grammatical comments:

- Line 31: Because this is an inline citation, an "and" is needed between Germanischer Lloyd (2010); and Stammler et al. (2024).
  The citation is corrected accordingly.

- Table 2: I recommend variables Z and I be italicized like other variables.
  The variable Z is changed to italic.

- Line 95: A space is missing at ".Some".
  Space is added in line 95